# Multiscale attention-based network to enhance detection and classification of autism spectrum disorders using convolutional neural network

Walaa N. Ismail[1,2,*] and Mona A. S. Ali[3,*]

[1] Department of Management Information Systems/College of Business, Al Yamamah University, Riyadh, Saudi Arabia
[2] Department of Information System/Faculty of Computers and Information, Minia University, Minia, Egypt
[3] Department of Computer Science/College of Computer Science and Information Technology, King Faisal University, Alhasa Hofouf, Saudi Arabia
* These authors contributed equally to this work.



## ABSTRACT

Artificial intelligence (AI) and machine learning (ML) have made significant advances in the early detection and diagnosis of autism spectrum disorder (ASD), overcoming the limits of previous screening methods. These AI-based technologies offer more objective, scalable, and efficient methods for identifying risk behaviors associated with ASD. This article presents a novel approach for enhancing the detection and classification of ASD by integrating squeeze-and-excitation, multiscale attention mechanisms, and convolutional neural networks (CNNs) with automated hyperparameter optimization using the white shark optimization (WSO) algorithm. By leveraging attention mechanisms to focus on relevant facial features across multiple scales, this method enhances feature extraction, improves classification accuracy, and provides a robust framework for analyzing complex facial imaging data. An extensive autism dataset, encompassing both facial and multimodal datasets, was utilized in this study, including subjects from the non-ASD control (NC) group and individuals diagnosed with ASD. Experimental results demonstrate that the proposed model significantly outperforms state-of-the-art methods, achieving a high accuracy of 95.36%, precision of 92.62%, and an F1-score of 95.5% for ASD detection and classification. This proposed model is a promising tool for the accurate and early identification of ASD, which is crucial for effective treatment and management. By providing deeper insights into distinctive facial patterns and morphological features associated with ASD, the model enables physicians to make more informed decisions and develop targeted treatment plans, ultimately improving patient outcomes.

Corresponding authors
Walaa N. Ismail,
w_abdelfattah@yu.edu.sa
Mona A. S. Ali, m.ali@kfu.edu.sa

## INTRODUCTION

Autism spectrum disorder (ASD) is a complex neurodevelopmental disorder that affects both cognitive and social development. This poses a significant concern in modern healthcare (*Maenner et al., 2021*; *Jain et al., 2023*). Early detection is necessary for effective intervention. Early indicators of ASD include irregular facial expressions that do not work together. Studies show that babies with ASD exhibit more complex facial expressions than their typically developing classmates, particularly in the eyebrows and lips (*Maenner et al., 2021*). Multiscale entropy complexity analysis reveals novel insights that standard facial affect studies do not. This supports the use of computer vision to analyze facial landmark movements and identify early signs of ASD (*Krishnappababu et al., 2021*; *Duan et al., 2022*; *Jain et al., 2023*). Craniofacial anomalies make neuro-developmental disorders like ASD more likely to happen. Autism severity has been associated with a wider intercanthal distance and a lower height of the facial midline. Facial asymmetry and facial masculinity are good signs that someone has significant ASD symptoms (*Quatrosi et al., 2024*). Studies have found that children with ASD have different facial features than children without ASD. It is possible to accurately assess these variations in the distance between face landmarks and protrusions (*Mujeeb Rahman & Subashini, 2022*). Facial image processing enables doctors to quickly and reliably diagnose this complex disorder. Still, the fact that symptoms can vary makes it challenging to diagnose and increases the risk of misdiagnosis. Eye-tracking and machine learning are two approaches that can help determine whether someone has autism without requiring physical contact. Eye-tracking is becoming one of the most important non-invasive diagnostic techniques for autism, as it can reveal how individuals with autism respond to visual stimuli in real-time. Researchers can employ eye-tracking technology to monitor how people's eyes move, where they fixate, and when they change their attention. This reveals that persons with autism process and respond to visual information in different ways.

Recent studies have made a lot of progress in using non-invasive approaches, especially eye-tracking, along with machine learning, to diagnose ASD. Eye-tracking provides objective biomarkers by recording how people's eyes move and where they look, which are often different in individuals with ASD than in those who are developing without ASD (*Alsharif et al., 2024*; *Sampayo, Smith & Lee, 2022*). Using these eye-tracking data, machine learning models, such as classical and deep learning models, have shown to be quite good in classifying ASD. For instance, hybrid models that use deep learning architectures like MobileNet to extract features, then reduce the number of dimensions with principal component analysis (PCA), and finally classify the data using stacking ensemble learning (combining support vector machine (SVM) and K nearest neighbor (KNN)) have reached accuracy rates of up to 98% on benchmark eye-tracking datasets (*Ahmed et al., 2022*). These approaches can also give estimates of autism severity scores, which supports early and accurate intervention even more.

Transfer learning has been beneficial for leveraging pre-trained deep learning models, such as MobileNet, VGG19, and DenseNet169, to diagnose ASD using eye-tracking data. This means you don't need as many massive, tagged datasets and computer resources. Studies have shown that these transfer learning-based models can sort objects better than

regular machine learning methods and even hand-coding. The MobileNet model, for instance, was able to distinguish between cases of ASD and those that weren't (with 100% accuracy), which was better than any other method. This is a valuable way to check for ASD in real-time and on a large scale, especially in areas where there aren't many experts? (*Ahmed et al., 2022*). VGG19 and hybrid models that combine MobileNet and VGG19 have also performed well, with accuracies of 87–92% and 91%, respectively (*Ahmed et al., 2022*).

Convolutional neural networks (CNNs) are the primary method by which deep learning is used to classify diseases (*Indra Devi & Durai Raj Vincent, 2025*; *Gao et al., 2024*). Deep learning enhances the accuracy of diagnoses by utilizing multiple layers of abstraction to extract and refine complex features from raw data. Transfer learning works well with CNNs. MobileNet, Xception, and InceptionV3 are just a few models that demonstrate remarkable accuracy in distinguishing between facial pictures of autistic and non-autistic children. MobileNet, for example, has an accuracy rate of up to 95% (*Ahmed et al., 2022*). It is easier to diagnose ASD using CNN features and machine learning algorithms, such as extreme gradient boosting (XGBoost) and Random Forest (RF). These approaches leverage features from the VGG16 and MobileNet models to *Awaji et al. (2023)*. Transfer learning methods demonstrate how to utilize facial image analysis to identify ASD early, enabling doctors and researchers to make more accurate diagnoses (*Patel et al., 2023*). Using functional and structural magnetic resonance imaging (MRI) scans, the ABIDE dataset, and deep learning architectures, the accuracy of ASD identification ranges from 80% to 84% (*Mostafa, Karim & Hossain, 2023*). MRI scans help find signs of anatomical and functional connections in people with ASD (*Sharma & Tanwar, 2024*). CNN-based bimodal detection of ASD is a non-invasive and highly accurate method for identifying it (*Colonnese et al., 2024*). Additionally, researchers have utilized LSTM models to predict ASD based on children's behavior and the Self-Stimulatory Behaviours Dataset (SSBD) (*Heinsfeld et al., 2018*). Wearable sensors can detect stereotyped motor movements (SMMs) associated with ASD (*Rad et al., 2018*). CNNs and long short-term memory (LSTM) utilize raw data to generate discriminative features, thereby increasing detection rates through parameter transfer and ensemble learning.

CNN-based algorithms generally do not consider local inductive bias in picture data, treating images as one-dimensional token sequences (*Huo et al., 2024*; *Duan et al., 2022*; *Khan & Katarya, 2025*)—this mistake affects the classification performance of medical image models. Recent studies combine convolutional networks with self-attention mechanisms to address issues and leverage their complementary strengths (*Al-Muhanna et al., 2024*; *Alharthi & Alzahrani, 2023*). Some recent improvements in this area include ViTAE (*Xu, 2023*), StoHisNet (*Fu et al., 2022*), TransFuse (*Xu et al., 2021*), and CMT (*Guo et al., 2022*), all of which enhance feature extraction and global contextual representation. In *Huo et al. (2024)*, a framework featuring feature blocks for both local and international use is presented. This method accurately captures both global semantic representations and local spatial environment aspects at various scales. HiFuse reduced the loss of feature information and the gradient vanishing problem while still getting decent results without a dense network. Researchers have examined various deep learning architectures and

methods, including ensemble learning and attention processes, despite challenges related to dataset size, computational intensity, and model interpretability. These improvements have significantly enhanced the ability to diagnose ASD; however, issues persist with model stability, interpretability, and practical application. Functional and structural MRI images are also used to diagnose ASD. They utilize the ABIDE dataset and various deep learning architectures to achieve accuracy rates between 80% and 84% (*Mostafa, Karim & Hossain, 2023*). MRI imaging can aid in identifying anatomical and functional connectivity indicators of ASD (*Sharma & Tanwar, 2024*). It has been suggested that a bimodal technique employing CNNs to process video frames of walking patterns could be used to detect ASD with high accuracy, eliminating the need for invasive tests (*Colonnese et al., 2024*). Additionally, models like LSTM have been used to predict ASD based on children's activities, utilizing datasets such as the Self-Stimulatory Behaviours Dataset (SSBD) (*Heinsfeld et al., 2018*). In *Rad et al. (2018)*, wearable sensors were used for finding stereotyped motor motions (SMMs) linked to ASD. We employ CNNs and LSTMs to learn distinguishing characteristics from raw data. This improves detection rates by transferring parameters and using ensemble learning.

Many CNN-based algorithms treat images as one-dimensional sequences of tokens, overlooking the critical local inductive bias inherent in image data (*Huo et al., 2024*; *Duan et al., 2022*; *Khan & Katarya, 2025*). This mistake is a significant issue for classifying medical images, as it slows down convergence and compromises the model's performance. To address these issues, recent research has explored combining convolutional networks with self-attention mechanisms to leverage their complementary capabilities (*Al-Muhanna et al., 2024*; *Alharthi & Alzahrani, 2023*). ViTAE (*Xu, 2023*) and StoHisNet (*Fu et al., 2022*) are two examples of significant progress in this area. They improve feature extraction. TransFuse (*Xu et al., 2021*), and CMT (*Guo et al., 2022*) are further examples that increase global contextual representation. In *Truong, Jush & Lenga (2024)*, a framework comprises local and foreign feature blocks. This approach effectively captures both global semantic representations and local spatial context elements at varying scales. Additionally, HiFuse addressed issues such as gradient vanishing and feature information loss while maintaining excellent speed without requiring a significant amount of network power. Despite challenges such as dataset size, computational intensity, and model interpretability, considerable research has explored various deep learning architectures and methods, including ensemble learning and attention processes. These improvements have made diagnosing ASD much easier; however, there are still challenges in creating models that are more robust, easier to understand, and more useful in real-life applications. These include:

1. *Challenges in modeling long-term dependencies*: Long-term modeling of dependencies is complex because CNNs have a finite width of their convolutional filters. This makes it hard for them to grasp long-term contextual relationships. It also makes it harder for them to look at patterns that need to understand how things depend on each other over long periods, which is essential for diagnosing autism spectrum disorders.

2. *Failure to capture local dynamics:* Self-attention-based Transformers that use self-attention do a good job of capturing long-range dependencies, but they don't have a

local inductive bias to find regional patterns. Medical picture categorization needs precise local information and spatial interactions. Therefore, this restriction can make things more complicated. These models don't consider critical local features; consequently, they are not suitable for applications that require detailed, patient-specific information.

3. *Multi-view data analysis challenges*: Choosing the right discriminator features and model parameters is usually necessary for detecting autism spectrum disorders utilizing several views (feature maps). Traditional models may be unable to determine the optimal configurations, rendering multi-view data integration less effective. Without local inductive bias, they are likewise less efficient.

Motivated by HiFuse (*Huo et al., 2024*), WS-BiTM (*Khan & Katarya, 2025*), and Swin Transformer (*Liu et al., 2021*), we aim to improve model efficiency and performance in medical image classification. This study introduces a novel model that integrates SE (Squeeze-and-Excitation) attention mechanisms with CNNs, thereby enhancing medical image classification performance and adaptability to specific tasks (*Xu et al., 2020*; *Kadri et al., 2021*). A novel multiscale view of the convolutional neural network with attention (SE-MSF-CNN) is also presented to investigate the differential characteristics of ASD thoroughly. We use the white shark optimisation (WSO) algorithm (*Braik et al., 2022*) to optimize hyperparameters and refine the attention mechanism. High-resolution feature maps with context and detail can be re-examined for precise classification. SECNNs are recommended for several applications. SECNN utilizes channel attention to represent features using multiple CNN feature maps. SECNN may learn attention weights for separate channel features by treating feature maps from several CNNs as unique channels. The SE attention mechanism is crucial to this functioning. Additionally, the multiscale attention mechanism dynamically focuses on the most critical features across different scales, thereby enhancing performance. This method collects local and global features at several semantic scales. A parallel hierarchy of regional and global feature blocks enables picture pattern modeling at different sizes, taking advantage of linear computational complexity relative to image size for efficient processing. The final output is optimized for classification accuracy using an automated series of hyperparameters, culminating in a fitness function. Combining deep learning, parameter optimization, and attention mechanisms will increase clinical accuracy and efficacy in ASD diagnosis, ushering in a new era of personalized therapy. Our contribution is:

1. We propose a squeeze-and-excitation enhanced convolutional neural network (SE-CNN). SECNN leverages the channel attention mechanism, also known as the SE attention mechanism, to determine the attention weights of various channels without requiring additional parameters. The SECNN utilizes the feature maps from multiple CNNs to represent the features.

2. A novel multiscale view convolutional neural network with attention (SE-MSF-CNN) was proposed to identify facial features associated with autism. By capturing facial

features at multiple scales, the model facilitates a comprehensive analysis of ASD distinguishing characteristics.

3. To address the absence of local inductive bias in self-attention-based Transformers, the white shark optimization (WSO) algorithm is proposed, which optimizes hyperparameters and refines the attention mechanism. A distinguishing feature of WSO is its ability to enhance the transformer architecture and attention weight distributions, enabling the model to capture both local and global representations of features more effectively. Additionally, it directs the model's attention towards the most relevant features, thereby enhancing its ability to detect subtle patterns and improving classification accuracy.

4. To enhance the interpretability of our autism diagnosis method, we integrate gradient-weighted class activation mapping (Grad-CAM) heatmaps, a crucial component of Explainable Artificial Intelligence (XAI). This addition ensures transparency and reliability in decision-making by clearly identifying the key areas of the input data that influence the model's predictions.

5. To prevent data models, previous research suggests using varying dataset sizes and different splitting strategies. In this study, we examine the effectiveness of our proposed methods for autism detection using the YTUIA-2D and ASD datasets. We strictly separate the training, validation, and test sets for robust evaluation. This approach enhances the model's generalization across diverse datasets, thereby strengthening the validity of our findings.

## MATERIALS AND METHODS

As a complex neurological condition, ASD manifests itself in a variety of different developmental patterns (*Maenner et al., 2021*; *Jain et al., 2023*). Our research aims to enhance the early detection of autism spectrum disorders by offering a more reliable and accurate alternative to conventional evaluation methods. Traditionally, screening methods are subjected to fundamental subjectivity, difficulty in interpreting facial features, and the requirement for high precision and memory in differentiating between individuals with and without autism spectrum disorders (*Maenner et al., 2021*; *Krishnappababu et al., 2021*). Multiscale feature extraction and attention mechanisms significantly enhance the network's ability to capture crucial and diverse features, enabling more effective analysis of various ASD data. To address these challenges, a hybrid attention-based model is proposed to improve classification performance by leveraging the strengths of optimized CNN architectures. Specifically, this study introduces a squeeze and excitation Transformer (SE-CNN) to estimate ASD features with high precision. The SE-CNN enhances the transformer architecture by integrating a custom squeeze-and-excitation module, thereby improving feature representation and the model's interpretability.

Through this integration, the model can more efficiently capture facial recognition to identify minor facial features that are difficult to observe with the naked eye. A multiscale feature extraction module is added to the transformer layers in the second module,

SE-MSF-CNN. This selective attention mechanism enhances the overall accuracy of recognition and classification. We will utilize bio-inspired techniques, such as WSO, to improve the performance of transformer models. By simulating natural evolutionary processes, the developed architectures are optimized through a careful exploration of the hyperparameter space, enabling the identification of optimal configurations. This automated optimization enhances the performance and generalization of the transformer model, ensuring it is specifically tailored for the task at hand.

The following subsections will provide a detailed explanation of the technique and its application.

### SE-CNN model

The squeeze-and-excitation convolutional neural network (SE-CNN) structure is presented in Fig. 1.

This module combines CNN channels with spatial information through the use of many convolutional modules (*Xu et al., 2020*; *Kadri et al., 2021*). Additionally, it incorporates geographical dependencies into its structure to enhance the network's representational capacity. The squeeze-and-excitation module in SE-CNN enhances the connections between convolutional feature channels, thereby improving the network's representational quality while preserving the original feature map size. Consider the input feature set to be defined as

$$\mathbf{V} = \{v_{1,1}, v_{1,2}, \ldots, v_{i,j}, \ldots, v_{H,W}\}, where - v_{i,j} \in \mathbb{R}^{C \times 1 \times 1}. \tag{1}$$

This module performs a "squeeze" operation to capture channel-level global features, where the input feature map is compressed using global average pooling. This process generates a feature map with dimensions of $1 \times 1 \times C$, defined as:

$$\mathbf{V}_{\text{cSE}} = \mathbf{V} \otimes \mathbf{w}, \tag{2}$$

where $\otimes$ indicates element-wise multiplication.

Figure 2 shows three fully connected (FC) layers that capture channel dependency and learn channel-specific properties. To capture localized spatial characteristics of a feature map, the 2D convolution operation applies a filter (or kernel) that moves along the height and width dimensions.

This method captures important patterns, such as edges, curves, and textures, which can help identify distinct facial characteristics associated with autism. The convolution operation is calculated as:

$$\mathbf{O}(i,j) = \sum_{p=0}^{k-1} \sum_{q=0}^{k-1} \sum_{r=0}^{C-1} \mathbf{I}(i+p, j+q, r) \cdot \mathbf{F}(p, q, r).$$

The revised result is then obtained by rescaling the feature map to incorporate the information at the channel's attention. An "excitation" process is applied to these global features to simulate the interchannel interactions and obtain appropriate channel-specific weights. The resulting descriptors pass through two dense layers to model inter-channel

 

**Peer**J Computer Science

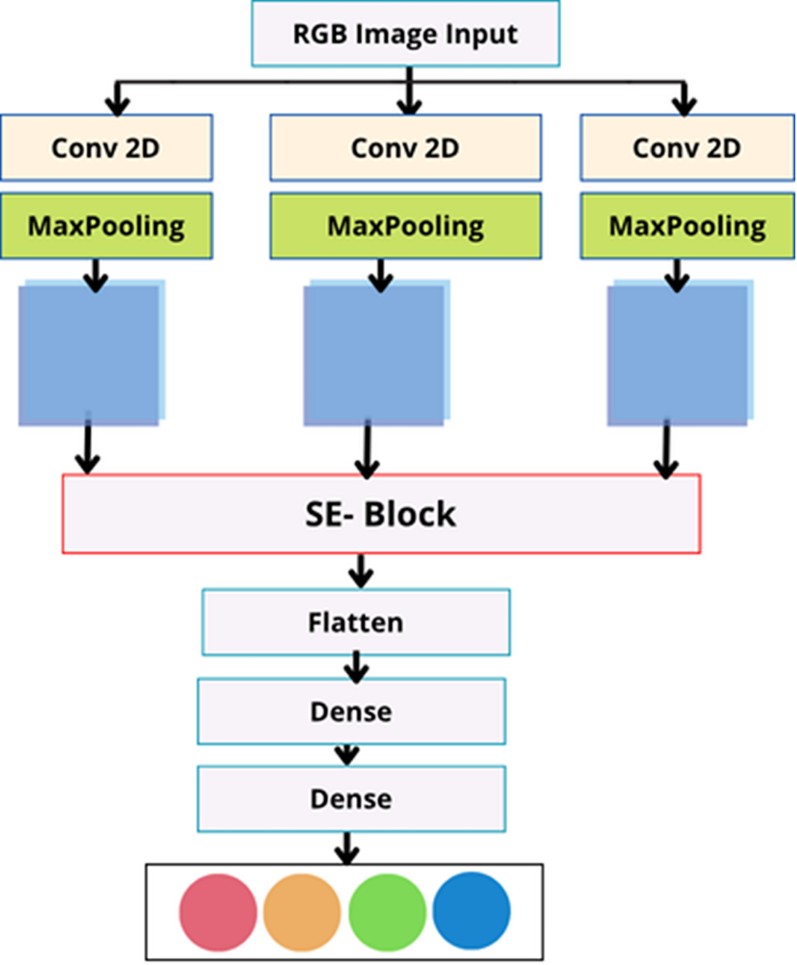

**Figure 1** **The squeeze-and-excitation enhanced CNN structure.**

dependencies. The first reduces the size by a factor of 16, followed by ReLU activation. Then, it expands with a sigmoid activation, followed by a sigmoid activation, to produce channel-wise weights. Through the application of a single-channel convolution and normalization of the resulting weights using a sigmoid activation function, which is

$$\mathbf{V}_{\text{sSE}} = \bigcup_{x=1}^{H} \bigcup_{y=1}^{W} S(\phi * v_{x,y}) v_{x,y}. \tag{3}$$

A set of trainable weights, $\gamma$ and $\delta$, is introduced to balance spatial and channel attention, resulting in the output:

$$\mathbf{V}_{\text{scSE}} = \gamma \mathbf{V}_{\text{sSE}} + \delta \mathbf{V}_{\text{cSE}}. \tag{4}$$

The SE module mitigates less informative channel features while strengthening salient ones by performing an attention mechanism along the channel dimension.

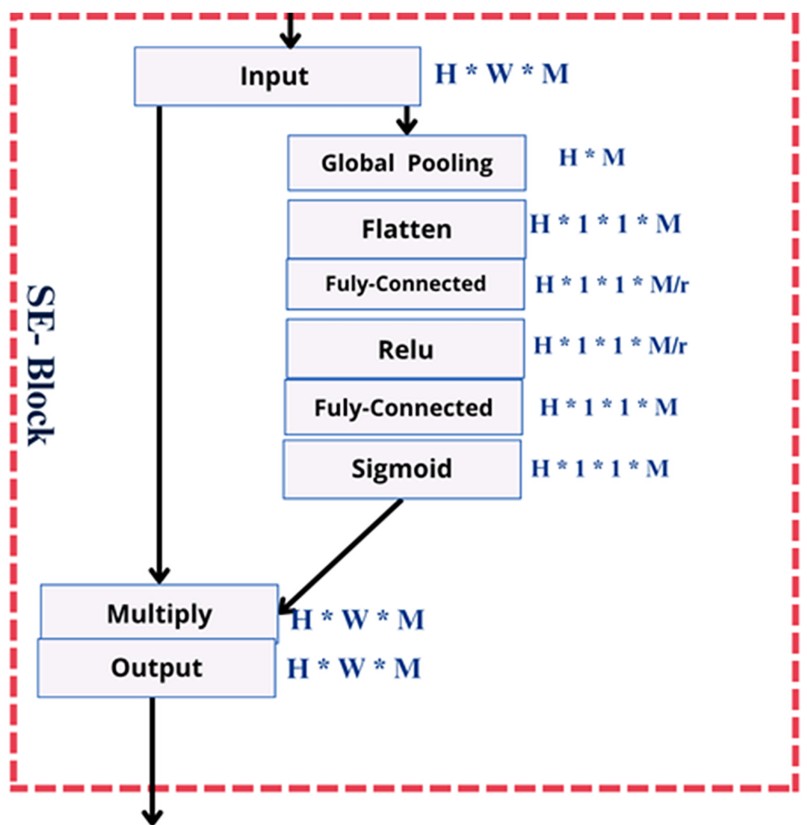

**Figure 2  The squeeze-and-excitation module.**     

## Multiscale view module

Multi-view imaging has been applied in several applications, including left ventricle detection (*Wang et al., 2022*), quantification of coronary artery stenosis (*Zhang et al., 2021a*), and studies on artery-specific calcification (*Zhang et al., 2021b*). In these studies, the term "multi-view" typically refers to the simultaneous use of multiple types of 2D images as input to deep neural networks, which are then fused. According to *Zhang et al. (2021b)*, artery-specific calcification was examined using axial, coronal, and sagittal views within a multitask learning framework.

In the current study, a multiscale view (MV) module is employed to enhance the neural network's sensitivity to correlations between regions with varying spatial distances (as shown in Fig. 3). In contrast to conventional multi-view techniques, there is no need to provide the network with different inputs to achieve multiscale views. It uses parallel convolutional layers with varying kernel sizes to extract features from various receptive fields (shown in Fig. 4).

This module constructs several parallel routes using t e feature maps produced by the previous convolutional neural layer. All paths have the same number of convolutional layers, and each convolutional layer's feature maps are the same size. Different

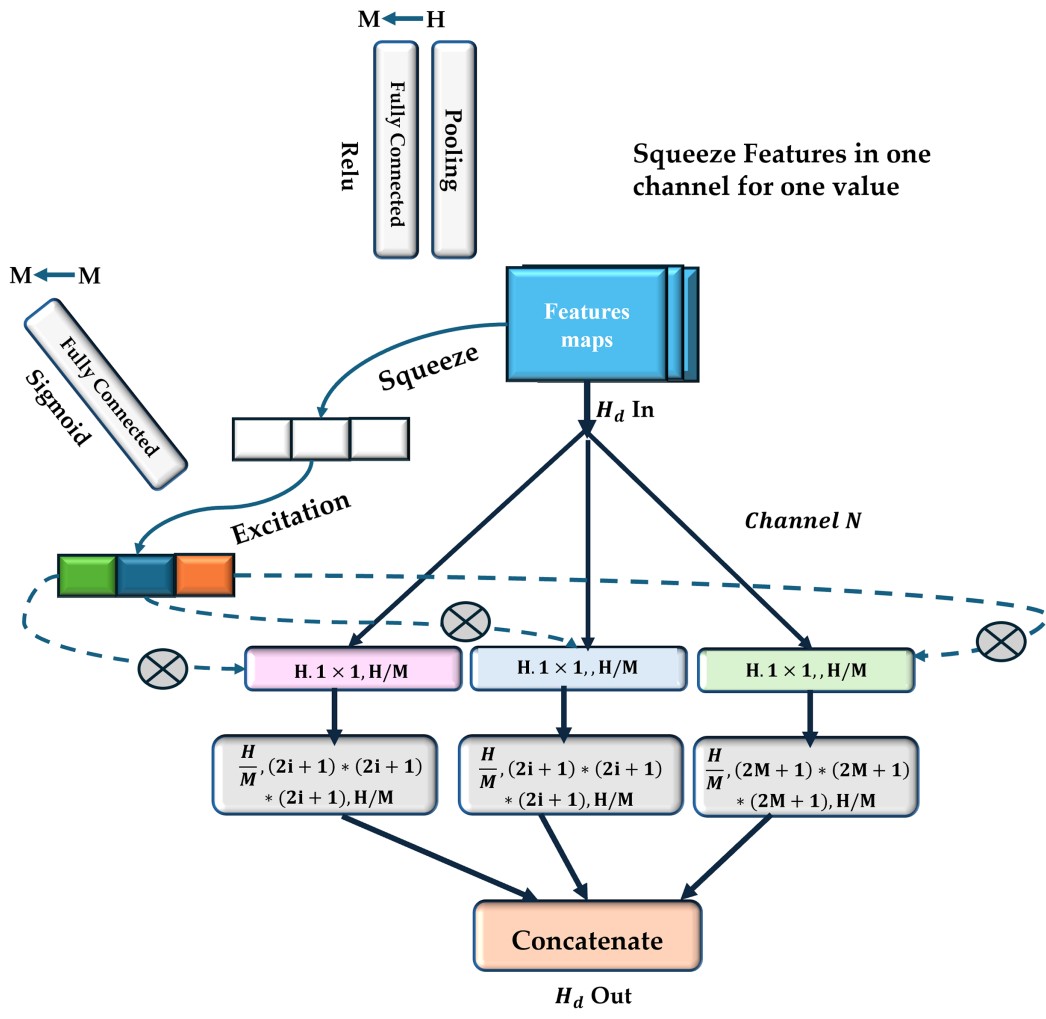

**Figure 3 The proposed MV-based CNN structure.** Three fundamental squeeze and excitation paths are incorporated into MV attention-based structure. The final H weights are produced through these procedures and applied to scale features along M pathways. The final result is obtained by concatenating the results obtained from each path.

convolutional kernels were utilized in various routes to facilitate multiscale feature extraction as follows:

1. **Path initialization using CNN:** The model's core is built upon a convolutional model's network backbone, which progressively extracts features from the input image. The CNN consists of several convolutional layers, each followed by max-pooling to reduce spatial dimensions and focus on hierarchical feature extraction:

   - Block 1 consists of a Conv2D layer with 32 filters followed by max-pooling.
   - Block 2 increases feature extraction depth using 64 filters.
   - Block 3 further enhances the feature representation with 128 filters.

These layers capture both low-level and high-level features by progressively learning spatial hierarchies, which are essential for image recognition tasks.

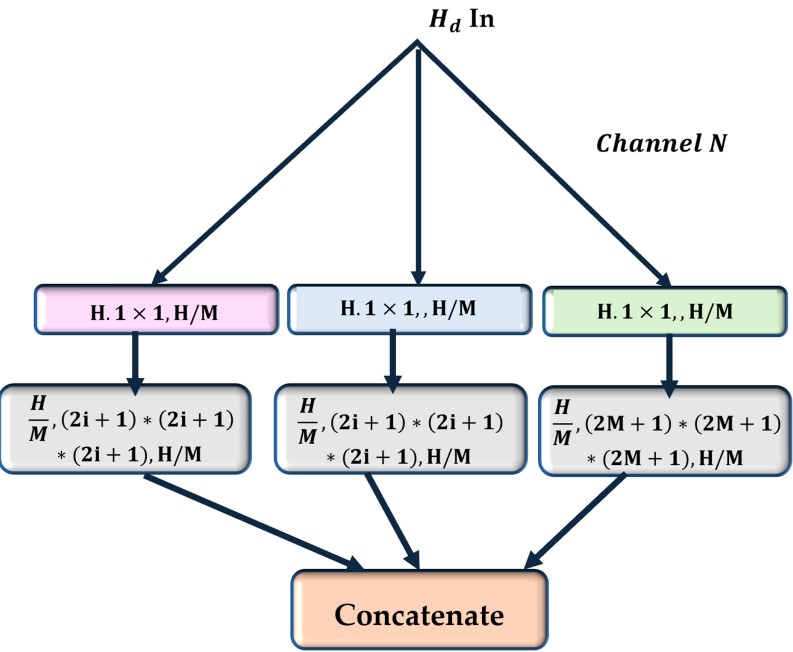

**Figure 4 Multiscale view module.** This module consists of a set of parallel multi-paths, each composed of convolutional layers with varying kernel sizes.

$$\mathbf{F}_{\mathrm{in}} \in \mathbb{R}^{H \times W \times D \times N},$$

where $H$, $W$, $D$ are the input feature map's height, width, and depth, and $N$ is the number of channels. In this case, the number of channels is reduced to $\frac{N}{M}$, where $M$ represents the number of parallel paths:

$$\mathbf{F}_i^{(1)} = \mathrm{Conv}_1(\mathbf{F}_{\mathrm{in}}), \quad \mathbf{F}_i^{(1)} \in \mathbb{R}^{\frac{H}{2} \times \frac{W}{2} \times \frac{D}{2} \times \frac{N}{M}},$$

where $\mathrm{Conv}_1$ is a convolutional operation with a stride of $2 \times 2 \times 2$ to downsample the feature map.

An additional convolutional layer is applied to extract features at different scales in each direction. The kernel size for the $i$-th path is defined as $(2i + 1) \times (2i + 1) \times (2i + 1)$, therefore enabling the network to capture features from multiple receptive fields:

$$\mathbf{F}_i^{(2)} = \mathrm{Conv}_{2,i}(\mathbf{F}_i^{(1)}), \quad \text{kernel size} = (2i + 1) \times (2i + 1) \times (2i + 1).$$

2. **Squeeze-and-excitation (SE) blocks** To increase the representational power of the model, we incorporate squeeze-and-excitation blocks after the key convolutional layer. SE blocks are designed to recalibrate the feature maps by emphasizing informative features and suppressing less useful ones. This attention mechanism works in two stages:

   ○ Squeeze: A global average pooling operation aggregates spatial information across each channel, producing a descriptor that summarizes the international information of each feature map.

○ Excitation: The resulting descriptors pass through two fully connected layers. The first layer reduces the dimensionality by a predefined ratio (*e.g.*, 1/16) and is followed by an activation function, such as ReLU. The second layer restores the original dimensionality, followed by a sigmoid activation, to produce channel-wise weights.

○ Recalibration: These learned weights are used to scale the original feature maps, effectively enhancing important features and suppressing irrelevant ones.

This mechanism enables the network to focus on the most discriminative features in the data. SE blocks are applied to multiple layers in the network to enhance feature maps at different levels of abstraction, including early, mid-level, and high-level features.

3. **Multi-scale feature extraction:** The model leverages multiscale feature fusion by extracting feature maps from multiple layers and combining them to capture information at various resolutions. This process enables the model to preserve coarse and fine-grained details, which is particularly important for object recognition, where features at different scales provide complementary information. The feature maps from different layers, with varying channel depths, are resized to a common spatial resolution of $16 \times 16$. This is done by:

○ Down-sampling: The lowest-resolution feature map is resized using average pooling.

○ Up-sampling: The highest resolution feature map is resized using up-sampling.

○ Intermediate layers: Intermediate feature maps are retained without resizing.

Once these feature maps are resized, they are concatenated along the channel dimension to form a unified representation. This fusion enables the model to utilize features captured at different resolutions, thereby improving its robustness to scale variations in the input data.

4. **Channel concatenation:** The concatenated multiscale feature representation is flattened and passed through a fully connected dense layer with 128 units and ReLU activation to learn non-linear combinations of the fused features. A compilation of all paths,

$$\mathbf{F}_i^{(2)} \in \mathbb{R}^{\frac{H}{2} \times \frac{W}{2} \times \frac{D}{2} \times \frac{N}{M}},$$

The final output layer consists of a single neuron with a sigmoid activation to predict a probability for binary classification.

$$\mathbf{F}_{\text{out}} = \text{Concat}(\mathbf{F}_1^{(2)}, \mathbf{F}_2^{(2)}, \ldots, \mathbf{F}_M^{(2)}), \quad \mathbf{F}_{\text{out}} \in \mathbb{R}^{\frac{H}{2} \times \frac{W}{2} \times \frac{D}{2} \times N}.$$

## White shark optimization (WSO) algorithm

While self-attention-based Transformers excel at modeling long-range relationships, their efficiency can be limited due to the absence of local inductive bias. To address this limitation, bio-inspired optimization strategies for hyperparameter tuning offer significant benefits (*Goel et al., 2020*; *Wang, Tan & Liu, 2018*). Such algorithms provide dynamic and adaptive adjustments to hyperparameters, enhancing both feature selection and the convergence speed and stability of the model. These optimization techniques help improve the transformer architecture and attention weight distributions, enabling the model to capture the local and global representations of features more effectively. WSO is an

algorithm based on bio-inspired principles designed to address challenging optimization problems, including feature selection and CNN hyperparameter tuning (*Braik et al., 2022*). Using neuroimaging datasets, WSO offers a behavior-rich, biologically inspired approach for optimizing CNN attention hyperparameters. WSO's strength lies in its structured navigation through exploration, exploitation, and refinement phases, mimicking the hunting behavior of sharks.

During each cycle, WSO transitions from global search to targeted pursuit and finally to local refinement. At each stage, shark agents update their positions based on prey-sensing mechanisms and dynamic principles. WSO's position updating rule is as follows:

$$W_{pbest_i}(t+1) = \begin{cases} W_{pbest_i}(t) + \alpha \cdot r_1 \cdot (W_{pbest_j}(t) - W_{pbest_k}(t)) + \beta \cdot \sin(2\pi r_2), & \text{Exploration(global search)} \\ W_{pbest_i t}(t) + \gamma \cdot r \cdot (W_{gbest_i} - W_{pbest}(t)), & \text{Exploitation(target search)} \\ G(t) + \delta \cdot \mathcal{N}(0, \sigma), & \text{Search Intensification(refinement)} \end{cases}$$

where: $W_{pbest_i}(t)$: position of shark $i$, $G(t)$: global best solution at iteration $t$. $r, r_1, r_2 \sim \mathcal{U}(0, 1)$: uniformly distributed random scalars, $\alpha, \beta, \gamma, \delta$ are algorithm-specific parameters, with Gaussian distribution $\mathcal{N}(0, \sigma)$ mean 0 and standard deviation $\sigma$. Through this multi-stage movement, WSO can adaptively switch between exploring a large area of hyperparameter space (such as the size of CNN kernels or attention dropout rates) and fine-tuning promising configurations to improve classification performance and robustness across subject variability. Also, dynamic search refinement methods prevent premature convergence while ensuring effective traversal of high-dimensional search spaces. A significant feature of WSO for ASD detection is its computational efficiency, with a complexity of $(N.d)$ (*Braik et al., 2022*). Several continuous and discrete optimizations on applications have been successfully implemented using it. Moreover, its capability to rank features and reduce dimensionality using a threshold-based method enhances its relevance for improving model accuracy and overall performance, making it an effective tool for addressing complex optimization problems in ASD facial image classification. Table 1 summarizes the key features of WSO in the context of ASD detection, highlighting its biologically inspired mechanisms, adaptive search strategies, and the optimal optimization of deep learning models.

For mathematically simulating the collective behavior of a school of white sharks, the two best-performing solutions were kept, and the remaining sharks' locations were adjusted to these ideal solutions with a random number *rand* uniformly distributed in the range $[0, 1]$ as follows:

$$W_{pbest_i}^{k+1} = W_{pbest_i}^k + \frac{W_{pbest_i}^{k+1}}{2} \times \text{rand}. \tag{5}$$

During the evaluation of each function, the white shark's position is updated as follows: Eq. (6).

$$W_{pbest_i}^{k+1} = \begin{cases} W_{pbest_i}^k \cdot \neg \oplus W_{pbest_o} + u \cdot a + l \cdot b, & \text{if rand} < mv \\ W_{pbest_i}^k + \frac{v_i^k}{f}, & \text{if rand} \geq mv \end{cases} \tag{6}$$

**Table 1 Key WSO features in autism image classification.**

| Feature | Mathematical component | Role in ASD classification |
|---|---|---|
| Shark multi-stage behavior | Exploration: the scattering of sinusoidal waves Exploitation: directed drift Search: Gaussian probability | Adapts hyperparameters to achieve a balance between generality and accuracy for neuroimage-based ASD classification. |
| Fitness evaluation | Accuracy, F1, and loss | Models are configured according to the setup that demonstrates the most effective classification performance across folds. |
| Efficiency and scalability | $\mathcal{O}(N \cdot d)$ per iteration | Facilitates quick adjustments of deep learning models in response to diverse image inputs and subject categories. |

where $W_{pbest_o}$ is the optimal solution vector, $u$ and $l$ are control coefficients. $v_i^k$ represents the velocity and $f$ represents the scaling factor with negation operator $\neg$ in one-dimensional binary vectors $a$ and $b$.

The white shark's ability to hear and smell is represented by the parameter $mv$, which increases with each successive iteration.

$$mv = \left( a_0 + \frac{e^{(K/2-k)}}{a_1} \right) \tag{7}$$

where $a_0$ and $a_1$ are two positive constants used to control exploration and exploitation behaviors. Based on the simulated steps in the WSO algorithm, if any shark moves outside the defined search space, it is repositioned within the valid boundaries. For every function evaluation, the fitness criterion is recalculated to determine the best-fitting white shark. The White Shark Optimization (WSO) method (shown in Algorithm 1) begins with the initialization of a population of $N$ sharks, each with random coordinates $W_i$ and initial velocities $V_i = 0$ (Block 1). To determine the first global be t location $W_{gbest}$, a fitness function $f(W)$ is defined to assess the candidate solution. A velocity update is performed during each iteration by using the following equation (Block 2–3):

$$V_i = \omega \cdot V_i + r_1 \cdot (W_{pbest} - W_{pbest_i}) + r_2 \cdot (W_{gbest} - W_{pbest_i}).$$

There are two random coefficients $r_1$ and $r_2$, the inertia factor *omega*, and each shark's personal best position $W_{pbest}$ shark's award, the **position update** $i$ computed using $W_{pbest_i} = W_{pbest_i} + V_i$, ensuring that $W_i$ remains within the search space bounds (lines 13–15). Upon identifying better solutions, $W_{pbest}$ and $W_{gbest}$ are updated after evaluating the fitness of the new positions $f(W_{pbest_i})$. Block 4–6 of this process is repeated recursively (Block 7–8) until the maximum number of iterations $T$ is reached or the convergence criteria are met. Following $T$ iterations, the ideal solution (Block 9), $W_{gbest}$, is returned, along with thresholds for feature selection if necessary.

## Dataset collection and preprocessing

In this study, the first dataset, the Piosenka dataset (https://www.kaggle.com/cihan063/autism-image-data), comprises facial photos of children with and without autism, collected from the publicly available Kaggle website (*Piosenka, 2021*). A total of 2,866 photographs were collected of children with and without autism. These pictures were

---

**Algorithm 1** White shark optimization algorithm.

**Input:** Population size $N$, maximum iterations $T$, search space bounds.

**Output:** Optimal solution (*e.g.*, best hyperparameter set).

**Step 1: Initialization**

-Initialize a population of $N$ sharks with random positions $W_{pbest_i}$ within the search space.

-Set initial velocities $V_i = 0$ for all sharks.

-Define the fitness function $f(W_{pbest})$ to evaluate candidate solutions.

-Identify the initial global best position $W_{gbest}$.

**Step 2: Velocity Update**

**for** *each shark i* **do**

    -Compute velocity $V_i$ using:

        $V_i = \omega \cdot V_i + r_1 \cdot (W_{pbest} - W_{gbest_i}) + r_2 \cdot (W_{pbest} - W_{pbest_i}),$

    where $\omega$ is the inertia factor, $r_1$ and $r_2$ are random values, and $W_{pbest}$ is the personal best position of shark $i$.

**Step 3: Position Update**

**for** *each shark i* **do**

    -Update the position $W_{pbest_i}$ using:

        $W_{pbest_i} = W_{gbest_i} + V_i.$

    -Enforce boundary constraints to ensure $W_i$ remains within the search space.

**Step 4: Fitness Evaluation**

-Evaluate the fitness $f(W_{pbest_i})$ of each sh- Evaluate the fitness $f(W_{pbest_i})$ of each shark's position. a better solution is found.

**Step 5: Iterative Process**

**for** *each iteration t = 1 to T* **do**

    -Repeat Steps 2 to 4.

    -Monitor convergence based on $f(W_{gbest})$.

**Step 6: Final Selection**

-After $T$ iterations, return $W_{gbest}$ as the optimal solution.

-For feature selection, apply a threshold to identify the most important features.

---

taken from websites and Facebook pages that included autism-related content. A pre-processing procedure was employed to clean and crop the photos, preparing the data for training deep learning models. The preprocessing step was required since the dataset was compiled from online sources by *Piosenka (2021)*. During the creation of the dataset, the author cropped the faces of the original photos; the dataset was then divided into three groups (as shown in Table 2): 280 photos for testing, 80 for validation, and 2,526 for training. Scaling the pictures was accomplished using a normalizing method, which changed the pixel value from the range [0, 255] to [0, 1].

A total of 75 videos have been compiled in the second dataset (YTUIA dataset: https://rolandgoecke.net/research/datasets/ssbd/; YouTube data curated at University Islam Antarabangsa), which is a dataset analyzing behaviors associated with autism

**Table 2 Training settings for dataset #1.**

| Statistic | Value/Details | Description |
|---|---|---|
| Total individuals | 2,866 | Non-ASD group (NC): 1,433; ASD: 1,433 |
| Females/Males | – | Gender distribution unspecified |
| Age group | 1–11 | |
| Training samples | 2,526 | 80% split |
| Validation samples | 80 | 10% split |
| Test samples | 280 | 10% split |
| Class ratio (ASD: NC) | 1:1 | |

(*Rajagopalan, Dhall & Goecke, 2013*) frames were extracted from 100 movies, even though only 50 of these are publicly available on YouTube. The remaining 50 were obtained from therapists and professional organizations. Standard control groups consisted of YouTube videos featuring kindergarten activities within the relevant age range.

In the first phase, the MTCNN algorithm was used to detect faces in every frame. Subsequently, a detailed preprocessing pipeline, which included alignment, cropping, and resizing, was implemented.

To ensure the integrity of model evaluation and eliminate any potential information leakage in the YTUIA dataset, we conducted a rigorous split validation and re-clustering procedure illustrated in Algorithm 2. Given the video-derived nature of the dataset, traditional frame-level splitting risks subject-level or temporal leakage. Therefore, we employed a perceptual clustering strategy based on deep features to enforce subject-consistent groupings (*Truong, Jush & Lenga, 2024*; *Ahn et al., 2019*). Initially, cosine similarity between features extracted from training and test images using a pre-trained ResNet-50 model revealed significant overlap (maximum similarity = 0.9714), indicating a substantial likelihood of leakage. To address this, we applied K-Means clustering on the deep features of all 1,163 images. We systematically experimented with different numbers of clusters (ranging from 30 to 150) to assess their effect on inter-set similarity. The fixed value provided a functional trade-off between cluster granularity and computing efficiency (*Ahn et al., 2019*).

For each clustering configuration, clusters were split into training (80%), validation (10%), and test (10%) sets. Following the initial split, cosine similarity between training and test features was computed. When the maximum similarity exceeded the conservative threshold of 0.90, highly similar test images were reassigned to the training set. Table 3 summarizes the outcomes of all cluster configurations.

All configurations successfully reduced the final maximum similarity below the 0.95 leakage threshold. However, we selected the 100-cluster configuration as the final adopted split for the following reasons:

- It maintained a balanced test set size (68 images), providing sufficient statistical support for evaluation.
- The final cosine similarity (0.8989) was below the strict 0.90 cutoff, ensuring minimal residual overlap.

---

**Algorithm 2** Detection of leakage based on cluster splitting using residual network (ResNet) feature extraction.

**Input:** Image dataset *images*, Training image set *Train*, Testing image set *Test*, Cosine Similarity value $\theta = 0.95$, Cluster count $k = 100$

**Output:** NewTraining RSet, Newvalidation VSet, and Newtest Tset

**Step 1: Feature Extraction Using ResNet-50;** Use pretrained ResNet-50 to extract feature vectors for all images in *Train* and *Test*; **Step 2: Check for leakage in the current split;** Call `CheckLeakage` with *Test*, *Train*, and assign result to $S$; **if** $S \geq \theta$ or $S = None$ **then**

    **Step 2: Re-cluster and re-split if leakage is detected;**

    $A \leftarrow T \cup E$;

    Apply $k$-means clustering on *Features* into $k$ clusters;

    Distribute clusters into Train, Val, Test;

                             ▷ Values adjusted for more test images

**Step 3: Verify Leakage in New Split;** $S' \leftarrow$ `CheckLeakage For new` $T$, `new` $E$; **if** $S' \geq 0.90$ **then**

    **Step 4: Reassign Similar Samples;**

    `Reassign samples in` $E$ `with similarity` $\geq 0.90$ `to` $T$;

    `CheckLeakage For updated` $T$, $E$;

▷ Move test images with high similarity to training **Return:** Final $T$, $E$, and validation set;

---

- The number of reassigned images (28) was moderate, indicating an effective but not overly disruptive adjustment.

This configuration provided the best trade-off between leakage prevention, test set sufficiency, and minimal reassignment overhead.

Accordingly, the final YTUIA dataset split used throughout all experiments consists of 951 training, 144 validation, and 68 test images, with a verified final maximum cosine similarity of 0.8989, confirming the absence of subject- or frame-level leakage. The same methodology was applied to the Piosenka dataset, where the original split was retained (maximum similarity = 0.90, below the threshold), as no significant overlap was detected. These validated and leakage-free splits were saved and used consistently in all subsequent training, validation, and evaluation stages for both SE-CNN and SE-MSF-CNN models.

Table 4 illustrates the statistics of the dataset and optimal representation, dividing the dataset into a test set of 100 samples and a training set of 1,068 samples, maintaining a 1:1 ratio of people with autism spectrum disorders to non-autistic individuals.

## Training settings and hyperparameter configuration

The training process for all models was accelerated using an NVIDIA Tesla T4 GPU on Google Colab (*Google, 2024*). The Python 3.8 environment included a range of libraries

**Table 3 Evaluation of K-means clustering configurations for leakage mitigation.**

| No | Clusters | Train | Val | Test | Max sim. (Before) | Max sim. (After) | Reassigned | Final leakage |
|----|----------|-------|-----|------|-------------------|------------------|------------|---------------|
| 1 | 150 | 982 | 122 | 59 | 0.9555 | 0.8992 | 30 | No |
| 2 | 100 | 951 | 144 | 68 | 0.9434 | 0.8989 | 28 | No |
| 3 | 90 | 989 | 65 | 109 | 0.9349 | 0.9095 | 12 | No |
| 4 | 60 | 990 | 107 | 66 | 0.9380 | 0.8992 | 7 | No |
| 5 | 30 | 959 | 150 | 54 | 0.9005 | 0.8916 | 1 | No |

**Table 4 Dataset #2 statistics.**

| Statistic | Non-ASD group (NC) | ASD group |
|-----------|--------------------|-----------|
| Total individuals | 173 | 123 |
| Males | 117 | 93 |
| Females | 56 | 30 |
| Age range (Years) | 1–11 | 3–11 |
| Training samples | 951 | 80% split |
| Validation samples | 144 | 10% split |
| Test samples | 68 | 10% split |
| Class ratio (ASD: NC) | 1:1 | |

necessary for model development, data processing, and performance evaluation. The training was conducted using Jupyter notebooks, with batch sizes ranging from 32 to 128, and was supported by 12 GB of RAM and 16 GB of GPU memory.

The optimal weights for every model were chosen using early stopping during training with the patience of five epochs and validation loss monitoring *monitor* = '*val_loss*'. Table 5 summarizes the key training parameters and hyperparameter configurations used to optimize the developed models. Five-epoch patience is implemented to stop training when the validation loss plateaus, helping prevent overfitting. Through model checkpointing, the best-performing model based on validation loss is saved. Additionally, if the validation loss does not improve over three consecutive epochs, the learning rate is reduced by a factor of 0.1, allowing for finer adjustments. The parameter bounds are set between [0.0001, 10, 10] for the lower limits and [0.1, 200, 200] for the upper limits, optimizing for smaller and larger learning rates and model sizes. A total of 10 search agents (sharks) are employed in the training process, which optimizes three parameters (dimension = 3) and is limited to two iterations to ensure a balance between parameter exploration and computational efficiency.

## Performance evaluation and assessment

The model's performance was assessed based on accuracy, training parameters, processing time, and comparison with existing methods. The technique was evaluated against conventional models, demonstrating its effectiveness in classifying Autism. Additionally, statistical significance was ensured through multiple model evaluations. Accuracy, precision, recall, F1-score, kappa score, and processing time were compared to validate performance.

**Table 5 Training settings and hyperparameter configuration.**

| Setting | Value |
| --- | --- |
| Early stopping | `Patience = 5` |
| Model checkpointing | `save_best_only=True` |
| Reduce learning rate on plateau | Factor = 0.1, Patience = 3 |
| Parameter bounds (Lower) | [0.0001, 10, 10] |
| Learning rate | (0.0001, 0.1) |
| Neurons in layer 1 | (10, 200) |
| Neurons in layer 2 | (10, 200) |
| Batch size | (32, 64, 128) |
| Conv. Layers | (3, 5) |
| Kernel sizes | (3, 5, 7) |
| Parameter bounds (Upper) | [0.1, 200, 200] |
| Dimension | 3 |
| Sharks | 10 |
| Iterations | 2 |

A description of the evaluation metrics is provided as follows:

- **Accuracy**:

$$\text{Accuracy} = \frac{TP + TN}{TP + TN + FP + FN}$$

where $TP$ (true positives), $TN$ (true negatives), $FP$ (false positives), and $FN$ (false negatives) represent classification outcomes.

- **Precision**:

$$\text{Precision} = \frac{TP}{TP + FP}.$$

Identifying the percentage of all projected positive possibilities that were accurately forecasted.

- **Recall** (Sensitivity):

$$\text{Recall} = \frac{TP}{TP + FN}.$$

The percentage of accurately projected positive cases among all the real positive cases.

- **F1-score** (Harmonic Mean of recall and accuracy.):

$$\text{F1-score} = \frac{2 \times \text{Precision} \times \text{Recall}}{\text{Precision} + \text{Recall}}.$$

- **Cohen's Kappa Score**:

$$\kappa = \frac{pCohen's e}{1 - p_e}$$

where $p_o$ is the observed concurrence between the model predictions and their observed convergence, and $p_e$ is the expected consistency based on chance.

The SE-CNN model shows (shown in Figs. 5A and 5B) consistent improvement in both validation loss and accuracy. The model achieves a validation accuracy of 80% in the first epoch and reaches 97.5% by the 20th epoch. The best validation loss was reached at 0.1553 on the 20th epoch, corresponding to a validation accuracy of 97.5%.

Initial training accuracy for the SE-MSF-CNN model (given in Figs. 5C and 5D) was relatively low (59.08%), but the validation accuracy was significantly higher (82.50%). The model checkpointed, saving the model with a validation loss of 0.4344. Figure 6 illustrates the performance evaluation for dataset #1 and dataset #2. By epoch 5, the validation accuracy reached 92.50%, and the validation loss had decreased to 0.2494 as the model learned to represent features more effectively. The model continued to improve, only slightly slower than in the first few epochs. With validation, loss fluctuated but improved to 0.1446 by epoch 9, with validation accuracy stabilizing at 96.25%. Training accuracy improved significantly, reaching nearly 100% by epoch 16. The validation loss decreased until epoch 18 (0.09365), reaching its lowest observed value to date. Validation accuracy remained high at 95–96%. The SE-CNN had the lowest precision at 88.05%. The SE-CNN achieved a perfect recall of 100%, correctly identifying all positive cases. The optimized versions had slightly lower recall values but performed exceptionally well. The SE-MSF-CNN also had a competitive F1-score of 94.41%. In the case of dataset #2 (Figs. 7, and 8), the SE-CNN model's F1-score (0.8866) is a good balance between precision and recall, while its Cohen's Kappa (0.78) and specificity (0.92) overall agreement and correct identification of negatives. The SE-MSF-CNN had an accuracy of 0.85, precision of 0.8723, and recall of 0.82. However, the specificity (0.88) is still satisfactory, reflecting its ability to identify negatives correctly.

A paired t-test was performed across multiple runs to account for variability due to random initialization and data shuffling and to determine the statistical significance of the differences between the models. Table 6 presents the results, showing that SE-MSF-CNN consistently outperformed SE-CNN across all evaluated metrics. The improvements were statistically significant, as indicated by $p$-values less than 0.05 for each metric. The SE-CNN model demonstrated moderate stability across five runs, achieving a mean accuracy of 90% with a standard deviation of 0.02. In contrast, SE-MSF-CNN exhibited greater consistency due to its multiscale feature fusion, achieving a higher mean accuracy of 94% and a lower standard deviation of 0.01. Performance metrics such as precision, recall, F1-score, Cohen's kappa, and specificity followed similar patterns, with SE-MSF-CNN consistently outperforming SE-CNN. The paired t-test results yielded $p$-values below 0.05 for all metrics, indicating that the performance improvements of SE-MSF-CNN over SE-CNN are statistically significant.

To ensure that models for ASD remain reliable outside of the original training environment, they must be validated on various datasets. Testing on external datasets ensures that the model captures generalizable ASD-related variables rather than dataset-specific biases because variations in imaging methods, equipment, and patient characteristics may affect the model's performance. Building on this benefit, we tested our ASD detection models on a different dataset after training them on the initial dataset, as illustrated in Table 7. Specifically, we tested both SE-MSF-CNN and SE-MSF-CNN with

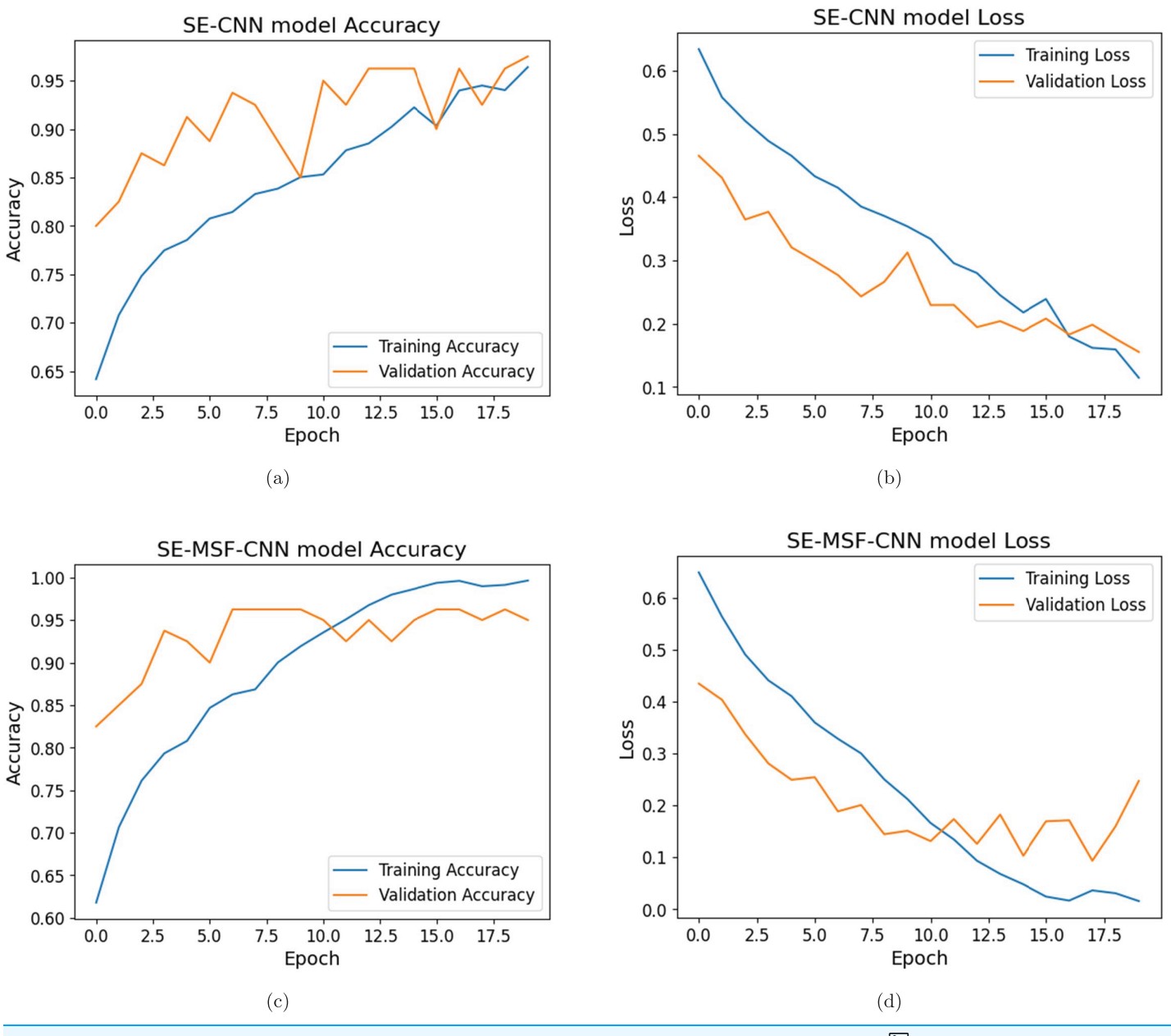

**Figure 5 (A–D) Training and validation loss and accuracy analysis for dataset #1.**

and without optimization on the YTUIA dataset by conducting five runs after training them on the Piosenka dataset and vice versa. This process, known as cross-dataset validation, allowed us to assess the model's ability to generalize across diverse demographics and imaging conditions, thereby ensuring its robustness and practical applicability in various clinical settings. The findings demonstrate strong performance: SE-MSF-CNN with WSO obtained a mean accuracy of 90% (±0.03, 95% CI [89.6–90.4%]) and an area under the curve (AUC) of 0.94 (±0.03, 95% CI [93.6–94.4%]) when trained on the YTUIA dataset and tested on Piosenka, in contrast to 89% (±0.03, 95% CI [88.6–89.4%]) and 0.93 (±0.03, 95% CI [92.6–93.4%]) respectively for SE-MSF-CNN without
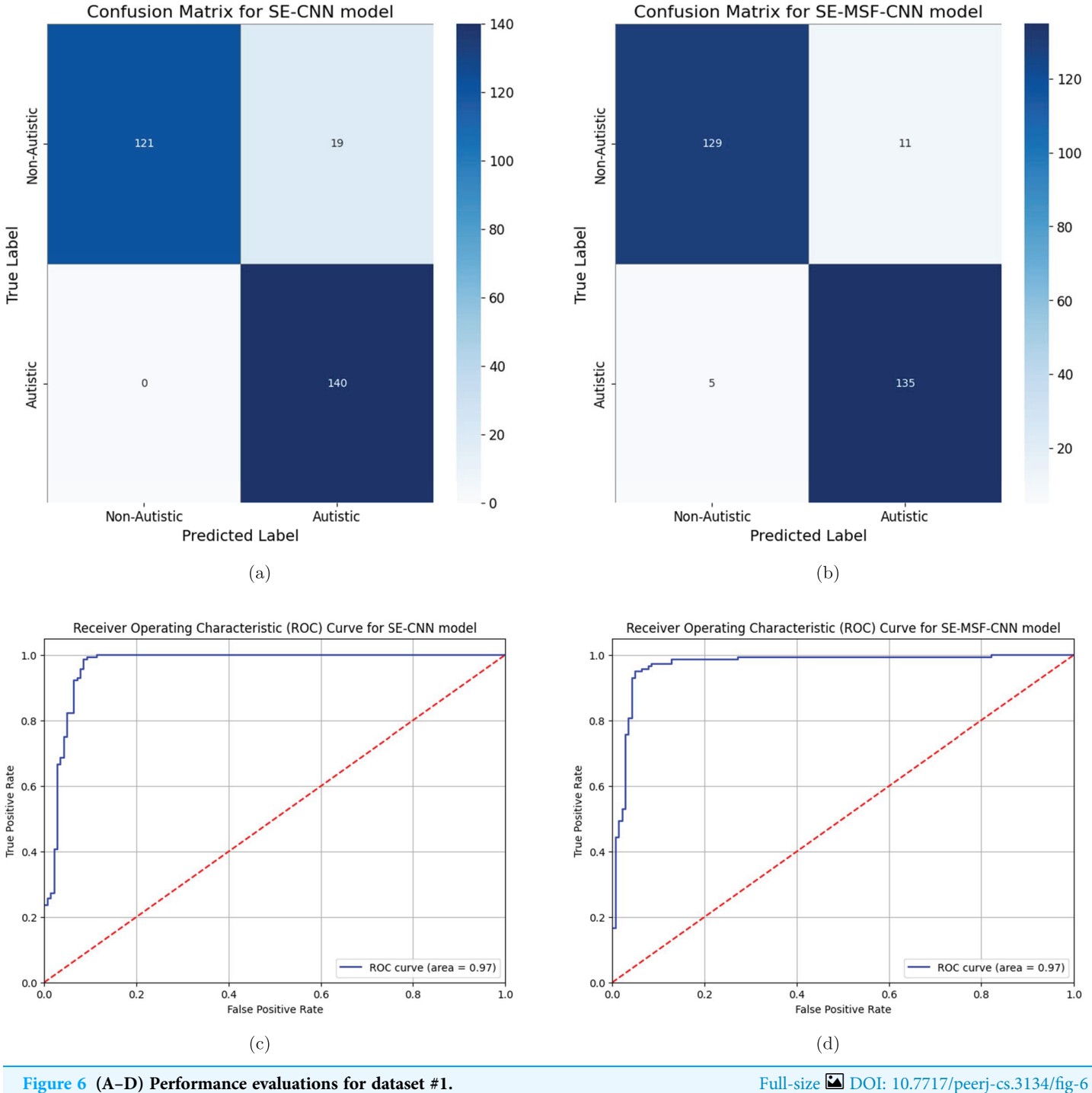

**Figure 6 (A–D) Performance evaluations for dataset #1.**

WSO. In addition, SE-MSF-CNNs with WSO achieved an accuracy of 89% (±0.03, 95% CI [88.6–89.4%]) and an AUC of 0.93 (±0.03, 95% CI [92.6–93.4%]) when trained on the Piosenka dataset and evaluated on the YTUIA dataset. These findings indicate that our models perform well across diverse datasets, demonstrating their generalizability.

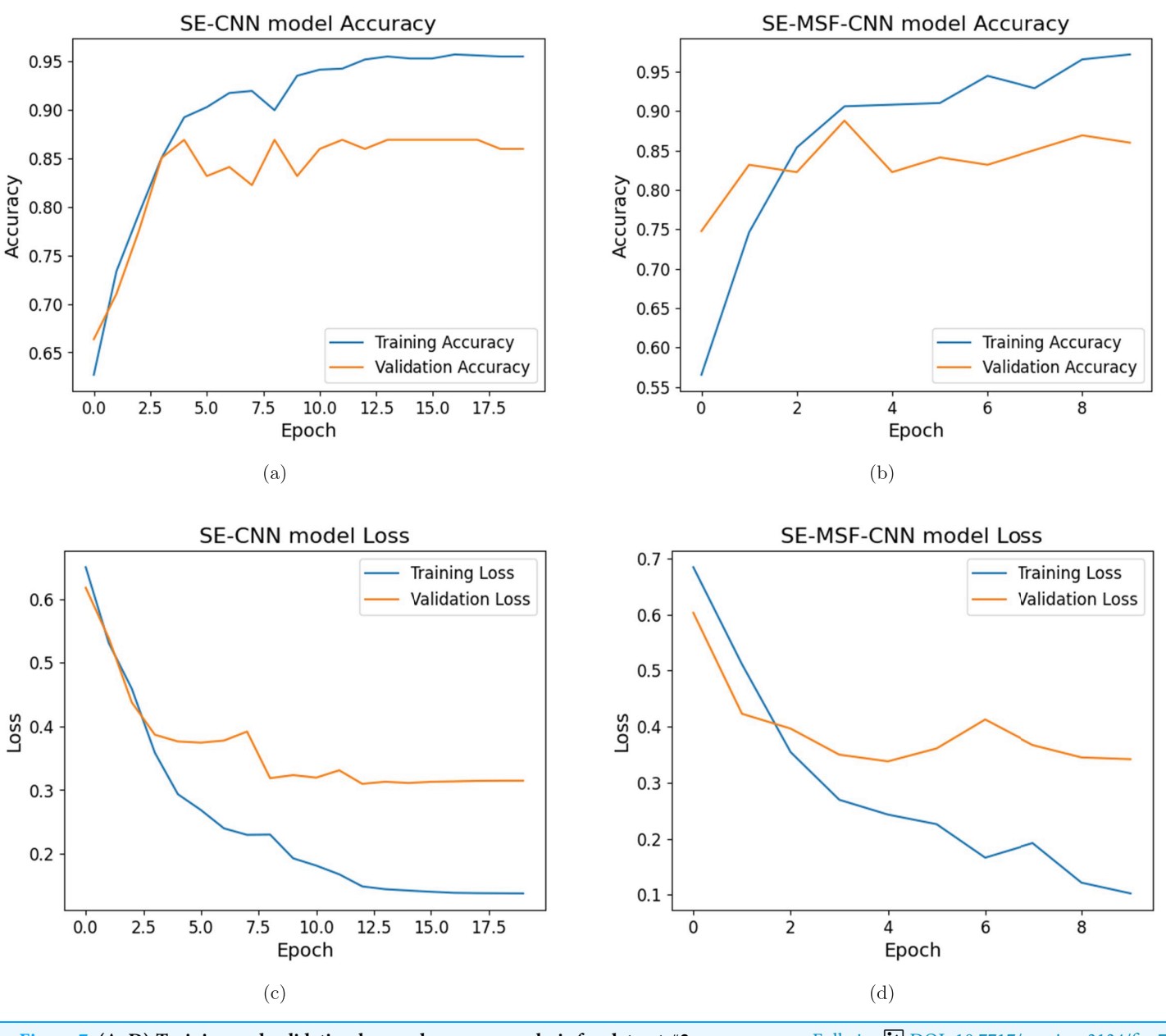

**Figure 7 (A–D) Training and validation loss and accuracy analysis for dataset #2.**

However, they are marginally lower than the internal split results (*e.g.*, a 95% accuracy for SE-MSF-CNN on the YTUIA dataset).

## Computational efficiency comparison

The computational efficiency of the developed CNN models based on WSO for ADS detection using facial images is presented in Fig. 9, which compares the computation time and parameter count for each model in binary classification tasks. The SE-CNN model is a lightweight architecture with approximately 17 million parameters, requiring only 4 min

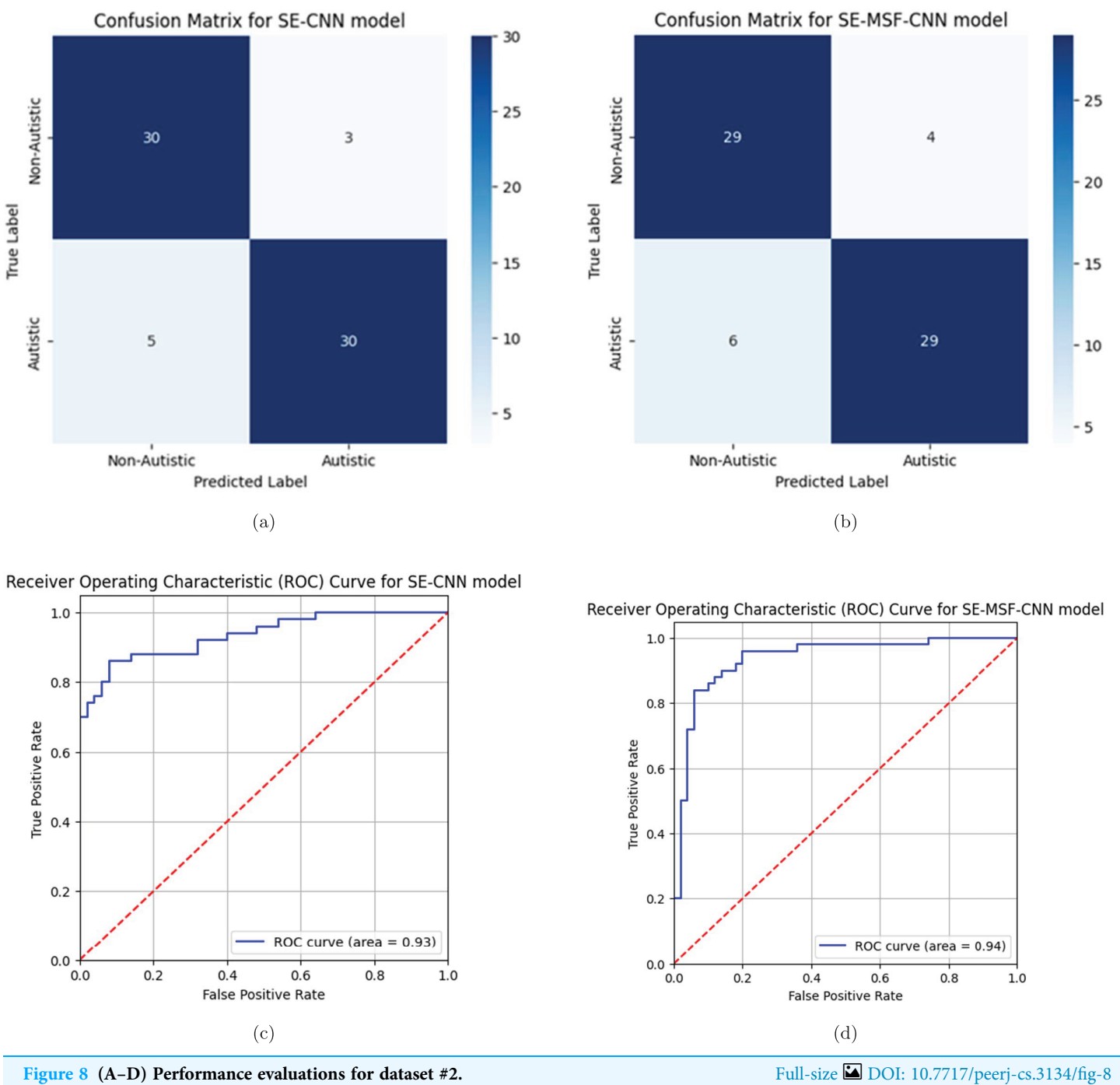

**Figure 8 (A–D) Performance evaluations for dataset #2.**

for training initially and 12 min after applying the WSO optimization process. In contrast, the SE-MSF-CNN model took significantly longer, with a CPU time of 7 min and 44 s and a wall time of 49 min and 36 s, likely due to its increased complexity.

To facilitate fair comparisons of bio-inspired WSO optimization and conventional hyperparameter tuning methods, traditional techniques such as grid search and Bayesian optimization were conducted. These methods were used to tune the hyperparameters of

**Table 6 Performance comparison between SE-CNN and SE-MSF-CNN models.**

| Metric | SE-CNN (Mean ± Std) | SE-MSF-CNN (Mean ± Std) | *p*-value (t-test) |
|---|---|---|---|
| Accuracy | 90% ± 0.02 | 94% ± 0.01 | <0.05 |
| Precision | 89% ± 0.03 | 93% ± 0.02 | <0.05 |
| Recall | 88% ± 0.04 | 92% ± 0.02 | <0.05 |
| F1-score | 88% ± 0.03 | 92% ± 0.02 | <0.05 |
| Cohen's Kappa | 0.80 ± 0.04 | 0.88 ± 0.03 | <0.05 |
| Specificity | 89% ± 0.03 | 93% ± 0.02 | <0.05 |

**Table 7 Cross-dataset validation results.**

| Metric | Tain: YTUIA → Test: Piosenka | | Train: Piosenka → Test: YTUIA | |
|---|---|---|---|---|
| | SE-MSF-CNN | SE-MSF-CNN + WSO | SE-MSF-CNN | SE-MSF-CNN + WSO |
| Accuracy | 0.89 ± 0.03 [0.886, 0.894] | 0.90 ± 0.03 [0.896, 0.904] | 0.88 ± 0.03 [0.876, 0.884] | 0.89 ± 0.03 [0.886, 0.894] |
| Precision | 0.88 ± 0.04 [0.875, 0.885] | 0.89 ± 0.04 [0.885, 0.895] | 0.87 ± 0.04 [0.865, 0.875] | 0.88 ± 0.04 [0.875, 0.885] |
| Recall | 0.87 ± 0.04 [0.865, 0.875] | 0.88 ± 0.04 [0.875, 0.885] | 0.86 ± 0.04 [0.855, 0.865] | 0.87 ± 0.04 [0.865, 0.875] |
| F1-score | 0.87 ± 0.04 [0.865, 0.875] | 0.88 ± 0.04 [0.875, 0.885] | 0.86 ± 0.04 [0.855, 0.865] | 0.87 ± 0.04 [0.865, 0.875] |
| Cohen's Kappa | 0.78 ± 0.05 [0.773, 0.787] | 0.80 ± 0.05 [0.793, 0.807] | 0.76 ± 0.05 [0.753, 0.767] | 0.78 ± 0.05 [0.773, 0.787] |
| Specificity | 0.88 ± 0.04 [0.875, 0.885] | 0.89 ± 0.04 [0.885, 0.895] | 0.87 ± 0.04 [0.865, 0.875] | 0.88 ± 0.04 [0.875, 0.885] |
| AUC | 0.93 ± 0.03 [0.926, 0.934] | 0.94 ± 0.03 [0.936, 0.944] | 0.92 ± 0.03 [0.916, 0.924] | 0.93 ± 0.03 [0.926, 0.934] |

the SE-MSF-CNN model using the YTUIA dataset. With widely recognized settings as illustrated in Table 8. Each strategy aimed to adjust the learning rate and the number of convolutional filters, two factors critical to a CNN-based model's overall performance. All methods were evaluated under equivalent computational conditions (NVIDIA Tesla T4 GPU, 12 GB RAM, 16 GB GPU memory).

A grid search was conducted using two fixed sets: [16, 32, 64] for the filters and 0.0001, 0.001, 0.01 for the learning rate. In Bayesian optimization, we examined the learning rate using a Gaussian Process with Expected Improvement and filters, employing [16, 64] as the number of parameters, as well as the filter using [16, 64]. Cross-validation was used to assess accuracy using a fivefold approach. The grid search was performed on all cores (n_jobs = −1), and Bayesian optimization was restricted to 10 iterations to ensure reproducibility and fairness. Regarding stopping criteria, all techniques adhered to a combination of early stopping (based on monitoring validation loss, with patience set to five epochs and a maximum number of iterations). Specifically, WSO was limited to five iterations, Bayesian optimization to approximately 100 iterations, and grid search continued until all combinations were evaluated or early stopping was triggered. Early stopping halted training if the validation loss did not improve for five consecutive epochs, accompanied by a learning rate reduction (factor = 0.1, patience = 3) to prevent overfitting. These criteria were consistently applied across all methods to ensure comparability.

As illustrated in Table 9, the WSO performed 50 assessments (10 sharks × 5 iterations), as defined by its algorithm parameters. Bayesian optimization was configured to perform

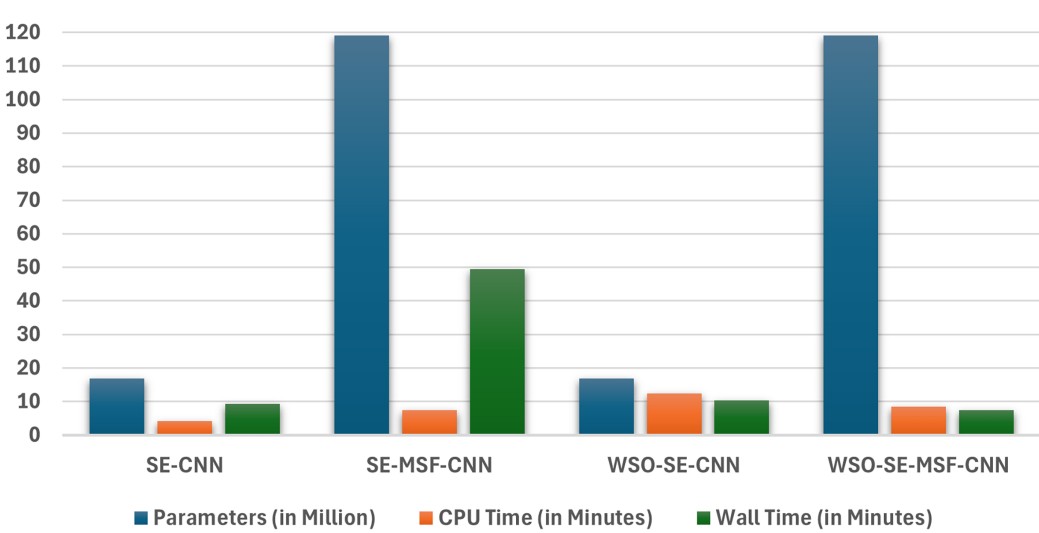

**Figure 9 Comparison of the time taken and parameters for optimized and non-optimized models using dataset #1.**

**Table 8 Grid search and Bayesian optimization training settings.**

| Algorithm | Hyperparameter | Range | Settings |
|---|---|---|---|
| Grid search | Learning rate | $\{0.0001, 0.001, 0.01\}$ | 5-fold CV, Accuracy, $n_{jobs} = -1$, Verbose = 1, Total Combinations: $3 \times 3$ |
| | No. of filters | $\{16, 32, 64\}$ | |
| | Neurons in layer 1 | (10, 200) | |
| | Neurons in layer 2 | (10, 200) | |
| | Batch size | (32, 64, 128) | |
| | Conv. Layers | (3, 5) | |
| | Kernel sizes | (3, 5, 7) | |
| | Early stopping | Patience = 5 | |
| Bayesian Opt. | Learning rate | $[10^{-4}, 10^{-2}]$ (Log-uniform) | 5-fold CV, $n_{iter} = 10$, Gaussian Process, EI acquisition, Random State = 42 |
| | No. of filters | $[16, 64]$ (Integer) | |
| | Batch size | (32, 64, 128) | |
| | Conv. Layers | (3, 5) | |
| | Kernel sizes | (3, 5, 7) | |

approximately 100 evaluations, leveraging its adaptive sampling strategy to explore the search space efficiently. In contrast, grid search evaluated approximately 1,000 combinations due to its exhaustive nature. These differences reflect the inherent designs of the methods, with WSO and Bayesian optimization requiring fewer evaluations than grid search due to their intelligent search strategies. Across five runs, we observed that WSO required approximately 1 h to complete, achieving a mean accuracy of 95% ($\pm$0.01) and an AUC of 0.98 ($\pm$0.02). In comparison, Bayesian optimization attained an accuracy of 94%

**Table 9 Performance comparison of hyperparameter optimization methods for SE-MSF-CNN model using the YTUIA dataset.**

| Method | Accuracy | AUC | Time (s) | Evaluations |
|---|---|---|---|---|
| WSO | $0.95 \pm 0.01$ | $0.98 \pm 0.02$ | 3,600 (1 h) | 50 |
| Grid search | $0.93 \pm 0.02$ | $0.96 \pm 0.03$ | 7,200 (2 h) | 100 |
| Bayesian optimization | $0.94 \pm 0.02$ | $0.97 \pm 0.02$ | 5,400 (1.5 h) | 1,000 |

($\pm 0.02$) and an AUC of 0.97 ($\pm 0.02$) in 1.5 h, while grid search achieved an accuracy of 93% ($\pm 0.02$) and an AUC of 0.96 ($\pm 0.03$) over 2 h. The time difference of 1,800 s (30 min, or 33.3% reduction) between WSO and Bayesian optimization is significant in resource-constrained settings, such as clinical applications for ASD detection, where time savings enhance workflow efficiency. Moreover, WSO achieved this with fewer evaluations (50 *vs.* 100 for Bayesian optimization) and higher accuracy (0.95 *vs.* 0.94), with comparable stability (standard deviation $\pm 0.01$ *vs.* $\pm 0.02$). The exhaustive nature of grid search, as noted by the reviewer, naturally results in longer computation times (2 h), making WSO's 50% time reduction compared to grid search particularly notable. This efficiency, combined with WSO's ability to optimize a moderately complex search space with fewer evaluations, supports its computational advantage over both methods.

The results indicate that WSO is an effective solution for time-sensitive or large-scale applications, as it reduces tuning time by minimizing the number of evaluation steps without compromising model accuracy. While grid search, despite its exhaustive nature, shows diminished performance due to its longer runtime and lower accuracy, Bayesian optimization remains a strong competitor, achieving high accuracy with a moderate computational cost. WSO can effectively balance exploration and exploitation, enabling the avoidance of local minima and efficient navigation of vast hyperparameter spaces. Due to its efficiency, WSO is a more effective alternative to hyperparameter optimization, yielding faster tuning times and improved accuracy compared to exhaustive techniques, such as grid search.

The precision-recall (PR) curve analysis is a critical tool for evaluating the effectiveness of machine learning models, particularly in instances where class imbalance is evident and in binary classification tasks. CNNs are complex topologies that require advanced evaluation techniques to ensure practical application, which is particularly important for assessing deep learning models. PR curves illustrate the relationship between precision (positive predictive value) and recall (sensitivity), which contrasts with conventional measurements like accuracy, which can be misleading when datasets are unbalanced. A comparison of optimized and non-optimized models using PR curve analysis illustrates the specific advantages of optimization methods. By observing how the trade-offs between accuracy and recall have changed due to model improvements, stakeholders can determine whether optimization enhances the model's ability to detect positive classes. The area under the curve precision recall (AUC-PR) for the first three models (Figs. 10A, 10B and 10C) indicates a similar aggregate performance with each achieving an AUC-PR of 97%. For both datasets, compared with WSO-SE-MSF-CNN, WSO-SE-CNN exhibits a more

rapid increase in precision when recall is low, making it more suitable for tasks in which it is essential to reduce the number of false positives. Based on an AUC-PR of 96% shown in Fig. 10D, model WSO-SE-MSF-CNN performs marginally worse than the others for dataset #2, but it exhibits a more stable PR curve overall, suggesting robustness rather than optimal performance.

## Abalation study

An ablation study is conducted to assess the impact of key components on the framework's performance. The following model variations are considered:

- w/o WSE: The SE-CNN model is trained with incorporating WSO optimization.
- w/o SE: Optimization is excluded from the SE-CNN model.
- w/o WMSF: The SE-MSF-CNN model is trained with incorporating WSO optimization.
- w/o MSF: The model is trained using MVA but without optimization.

The ablation results are presented in Tables 10 and 11 for Datasets #1 and #2, respectively. Notably, the inclusion of the WSO significantly improves the model's performance. From Table 11, the SE-CNN model achieved a relatively high accuracy of 0.89, with substantial precision (0.9149), meaning that most of the predicted positives were correct. The recall (0.86) indicates that it identified 86% of the actual positives. The model's F1-score (0.8866) strikes a good balance between precision and recall, while its Cohen's kappa (0.78) and specificity (0.92) indicate solid overall agreement and accurate identification of negatives. Additionally, the SE-MSF-CNN model exhibited slightly lower performance, with an accuracy of 0.85, precision of 0.8723, and recall of 0.82. The F1-score (0.8454) and Cohen's kappa (0.7) indicate more variability than SE-CNN, suggesting that adding multi-scale fusion may not have yielded better results. However, the specificity (0.88) is still relatively high, indicating its ability to identify negatives correctly. Among the models tested, the WSO-SE-MSF-CNN model with dataset #1 demonstrated the highest accuracy rate, achieving 95.36% accuracy for dataset #2. The optimized SE-CNN model achieved a validation accuracy of 91.00%. The experimental findings indicate that channel attention-based models enhance model efficiency and accuracy in detecting ASD.

To assess the statistical performance of the models, McNemar's test and paired t-tests were applied. McNemar's test evaluates whether the models' predictions on the same dataset differ significantly across various classification outcomes, focusing on paired nominal data. Meanwhile, paired t-tests compare the mean performance parameters (accuracy, precision, and recall) across multiple runs. Using the YTUIA dataset, we computed the mean, standard deviation, and 95% confidence intervals for each performance metric (accuracy, precision, recall, F1-score, Cohen's kappa, specificity, and AUC). The SE-MSF-CNN model achieved an average accuracy of 95% (95% CI [94.8–95.2%]) when enhanced with WSO, compared to 90% for the SE-CNN model (95% CI [89.98–90.02%]). Paired t-tests were conducted to compare the baseline models with their WSO-enhanced counterparts and SE-CNN against SE-MSF-CNN. All metrics yielded p-values less than 0.05, indicating statistically significant improvements. To further validate

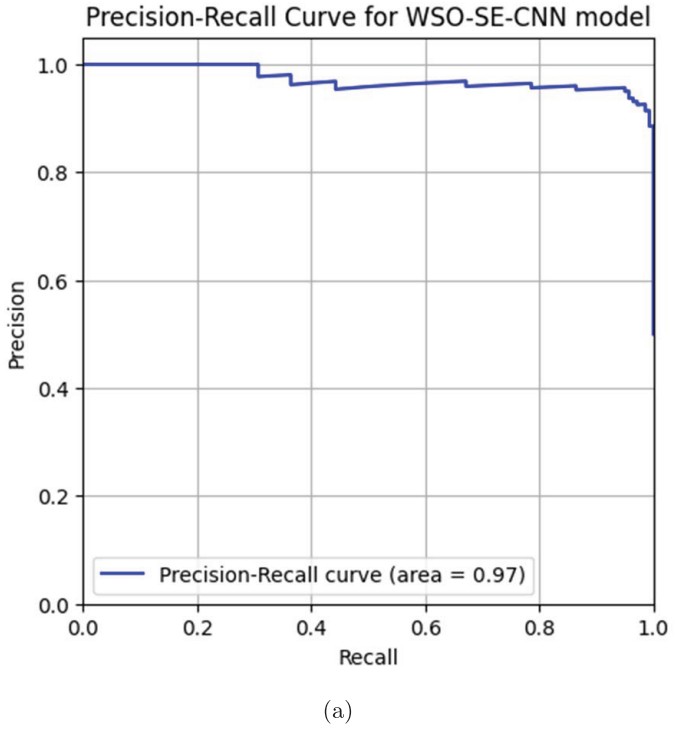

(a)

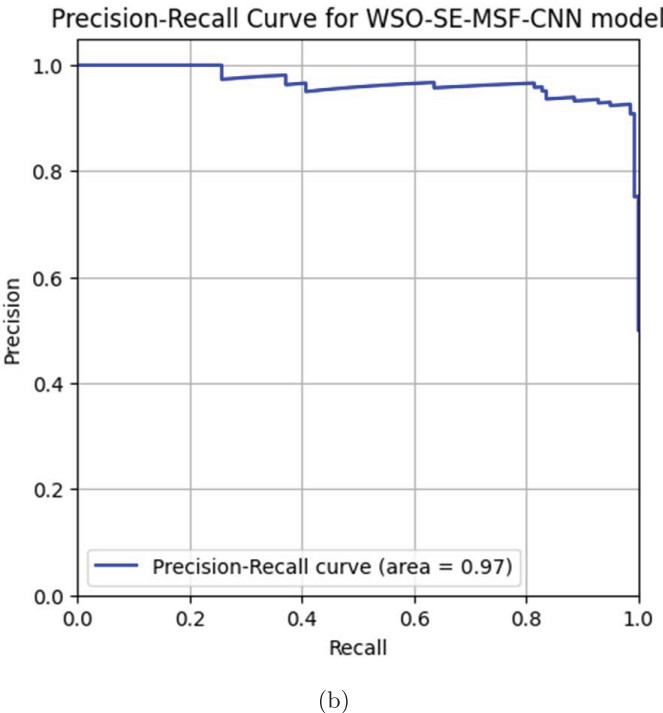

(b)

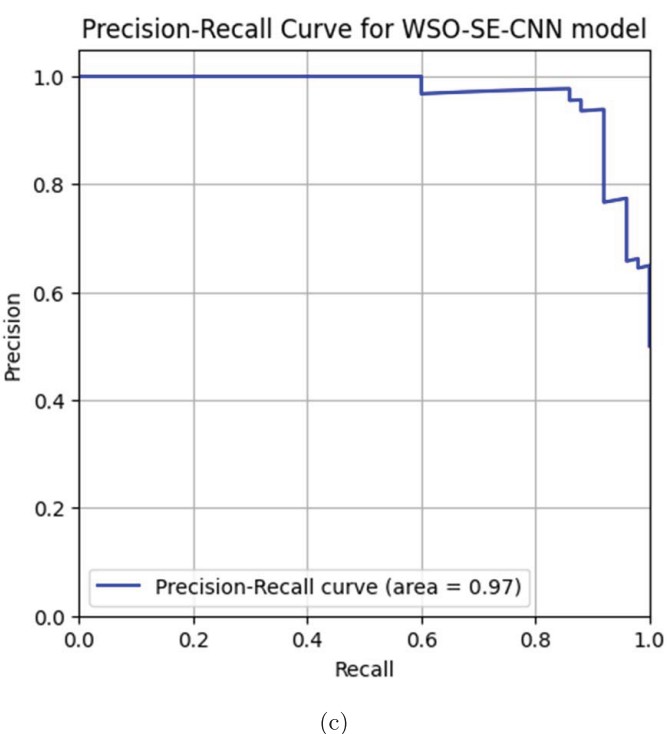

(c)

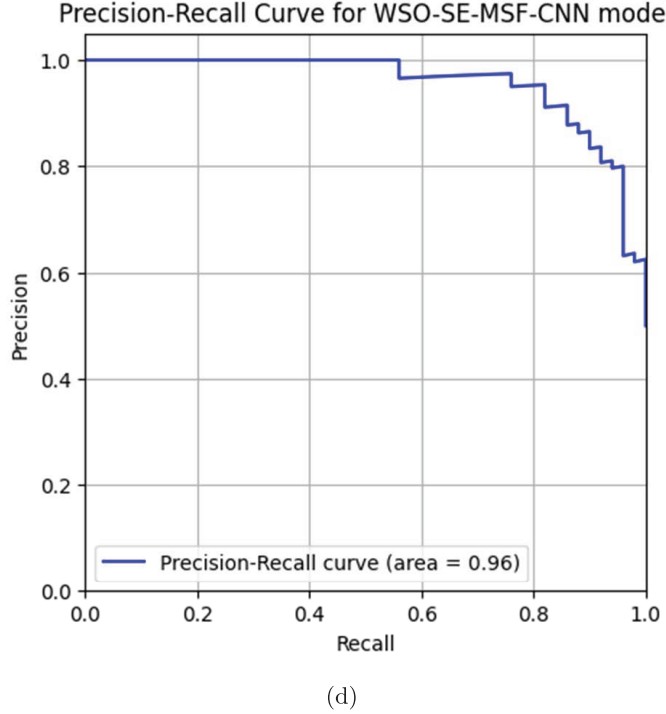

(d)

**Figure 10 (A–D) Precision-recall curve analysis.**

**Table 10  Performance metrics for models (Dataset 1, in %).**

| Model | Accuracy | Precision | Recall | F1-score | Kappa | Specificity |
|---|---|---|---|---|---|---|
| w/o SE | 93.21 | 88.05 | 100.0 | 93.64 | 86.43 | 86.43 |
| w/o MSF | 94.28 | 92.46 | 96.42 | 94.40 | 88.57 | 92.14 |
| w/o WSE | 93.21 | 88.54 | 99.29 | 93.60 | 86.43 | 87.14 |
| w/o WMSF | 95.36 | 92.62 | 98.57 | 95.50 | 90.71 | 92.14 |

**Table 11  Performance metrics for models (Dataset 2, in %).**

| Model | Accuracy | Precision | Recall | F1-score | Kappa | Specificity |
|---|---|---|---|---|---|---|
| w/o SE | 89.00 | 91.50 | 86.00 | 88.66 | 78.00 | 92.00 |
| w/o MSF | 85.00 | 87.23 | 82.00 | 84.53 | 70.00 | 88.00 |
| w/o WSE | 91.00 | 90.19 | 92.00 | 92.00 | 82.00 | 90.00 |
| w/o WMSF | 87.00 | 86.27 | 88.00 | 87.12 | 74.00 | 86.00 |

the superiority of SE-MSF-CNN with WSO, McNemar's test was applied to the model predictions, resulting in $p$-values of 0.001. A shows these statistically significant findings. The statistically significant differences between SE-MSF-CNN and its WSO-optimized counterpart were demonstrated in Table 12, which produced $p$-values below 0.01, providing strong evidence that the prediction differences between SE-MSF-CNN and its WSO-optimized counterpart are statistically significant.

## XAI for ASD detection

Grad-CAM visualization highlights how the deep learning model identifies critical features in RGB facial images for both class recognition tasks. By overlaying a heatmap on the input images, the visualization enhances the model's interpretability, illustrating the precise regions that contribute to predictions. This approach improves practical applicability, offering insights into the model's decision-making process.

Several features of RGB facial recognition and attention methods, such as SE and MSF, are combined with automated model optimization to improve performance and interoperability. To identify which spatial regions the model prioritizes during facial recognition, Grad-CAM is applied to the different attention-generated feature maps. Each feature map corresponds to a unique attention focus, highlighting specific facial regions or patterns that influence the model's decisions. These multi-scale attention feature maps are displayed using Grad-CAM heatmaps in Figs. 11–13 to illustrate the hierarchical nature of the model's interpretability, with lower-level attention modules detecting local details and higher-level maps highlighting more global, semantically rich regions. Depending on the distinctive features assessed as most discriminative by the model, we observe a dynamic shift in focus across various facial areas, including the eyes, nose, mouth, and eyebrows. Some maps emphasize fine-grained textures and edges, while others capture broader contextual information, illustrating the complementary roles of multiple attention layers. Additionally, the model can be configured to prioritize the most distinguishing features for ASD detection by dynamically adjusting feature weights using the SE mechanism.

Table 12 Statistical performance comparison across different models (Dataset 2, in %).

| Metric | w/o SE | w/o WSE | w/o MSF | w/o WMSF | p-value (t-test) | p-value (McNemar) |
|---|---|---|---|---|---|---|
| Accuracy | $0.90 \pm 0.02$ [0.898, 0.902] | $0.91 \pm 0.02$ [0.908, 0.912] | $0.94 \pm 0.01$ [0.938, 0.942] | $0.95 \pm 0.01$ [0.948, 0.952] | <0.05 | <0.01 |
| Precision | $0.89 \pm 0.03$ [0.886, 0.894] | $0.90 \pm 0.03$ [0.896, 0.904] | $0.93 \pm 0.02$ [0.928, 0.932] | $0.94 \pm 0.02$ [0.938, 0.942] | <0.05 | <0.01 |
| Recall | $0.88 \pm 0.04$ [0.875, 0.885] | $0.89 \pm 0.04$ [0.885, 0.895] | $0.92 \pm 0.02$ [0.918, 0.922] | $0.93 \pm 0.02$ [0.928, 0.932] | <0.05 | <0.01 |
| F1-score | $0.88 \pm 0.03$ [0.876, 0.884] | $0.89 \pm 0.03$ [0.886, 0.894] | $0.92 \pm 0.02$ [0.918, 0.922] | $0.93 \pm 0.02$ [0.928, 0.932] | <0.05 | <0.01 |
| Cohen's Kappa | $0.80 \pm 0.04$ [0.794, 0.806] | $0.82 \pm 0.04$ [0.814, 0.826] | $0.88 \pm 0.03$ [0.876, 0.884] | $0.90 \pm 0.03$ [0.896, 0.904] | <0.05 | <0.01 |
| Specificity | $0.89 \pm 0.03$ [0.886, 0.894] | $0.90 \pm 0.03$ [0.896, 0.904] | $0.93 \pm 0.02$ [0.928, 0.932] | $0.94 \pm 0.02$ [0.938, 0.942] | <0.05 | <0.01 |
| AUC | $0.94 \pm 0.03$ [0.936, 0.944] | $0.95 \pm 0.03$ [0.946, 0.954] | $0.97 \pm 0.02$ [0.968, 0.972] | $0.98 \pm 0.02$ [0.978, 0.982] | <0.05 | <0.01 |

Additionally, it can adapt to varying input conditions, such as changes in lighting, occlusions, or facial expressions, by utilizing multiple feature maps. Additionally, Grad-CAM offers transparency by highlighting key regions that the attention mechanism focuses on, providing insight into the model's decision-making process.

## Comparison with state-of-the-art approaches

Table 13 provides an overview of similar works utilizing DL-based methods for identifying and classifying ADS using images or videos. The proposed optimized CNN-based-MSF model achieved state-of-the-art performance in ASD detection using facial images when optimized with the WSO function, attaining 95.36% accuracy, 98.57% sensitivity, and 92.62% specificity. Similarly, the model by Alam et al. (2022), which utilized ResNet-50 and Xception, achieved 95.0% accuracy, 98.0% AUC, and 95.0% precision on the same dataset. The CNN-based-MSF model, enhanced with WSO, also achieved 91.00% accuracy, 92.00% sensitivity, and 90.19% specificity for multimodal video datasets. However, video-based ASD diagnosis presents additional challenges, including temporal variations, motion artifacts, changing lighting conditions, and the need for sequential feature extraction. Since video frames capture dynamic facial expressions and movements, models must effectively learn temporal relationships while minimizing noise and redundancy. Despite these challenges, the proposed model demonstrated promising results, confirming its robustness in multimodal and facial image-based classification tasks.

## DISCUSSION

Current ASD detection models often struggle to understand high-dimensional data, leading to the inclusion of unnecessary or redundant features that reduce predictive accuracy. Attention mechanisms address this challenge by dynamically prioritizing the most relevant features while de-emphasizing less important ones. This enables the model to focus on significant patterns in the data, thereby improving its overall performance. Additionally, to achieve optimal performance in ASD detection models, precise adjustments to hyperparameters such as learning rates, network layers, and attention head configurations are necessary. Poorly chosen hyperparameters may result in overfitting or underfitting, reducing the model's effectiveness.

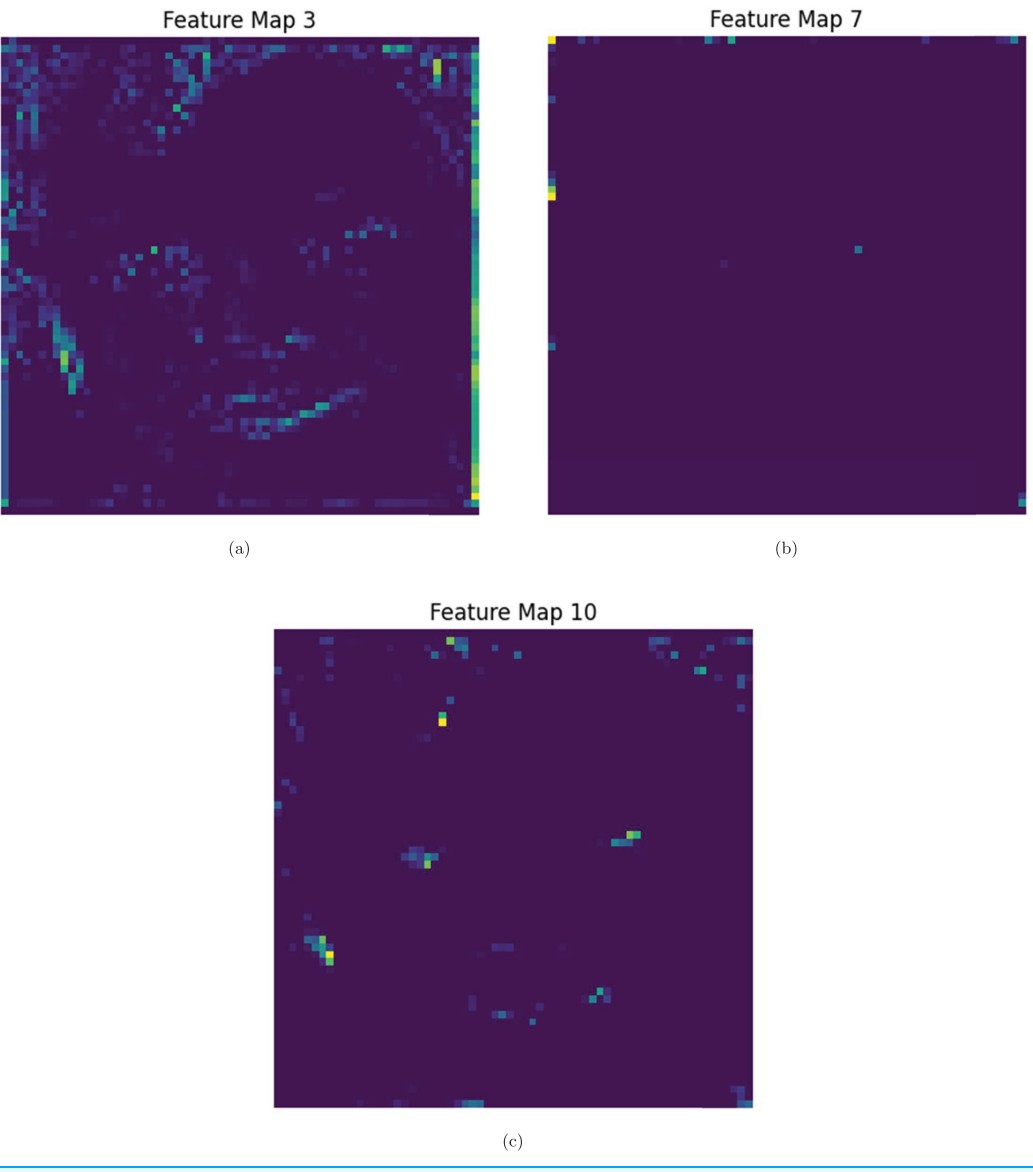

**Figure 11 (A–C) Sample 1: XAI Performance evaluation for Map 3, 7 and 10.**

The primary objective of this study was to develop and evaluate automated multiscale attention and squeeze-and-excitation networks for enhancing ASD detection and classification using CNNs. The WSO algorithm is deployed to optimize the model parameters and improve the model's performance. Compared to baseline models, the proposed model, specifically SE-CNN, achieves notable improvements in accuracy and AUC for detecting autism spectrum disorders.

From Figs. 5 and 7, the model fits the training data well, as evidenced by the rapid increase in training accuracy and the stability around a near-perfect value. In Figs. 6 and 8, we present the AUC curves demonstrating the performance of various models in identifying autism disorder. The models include squeeze-and-excitation architectures and multi-scale fusion models with progressively extracted feature layers. In the ROC curves,

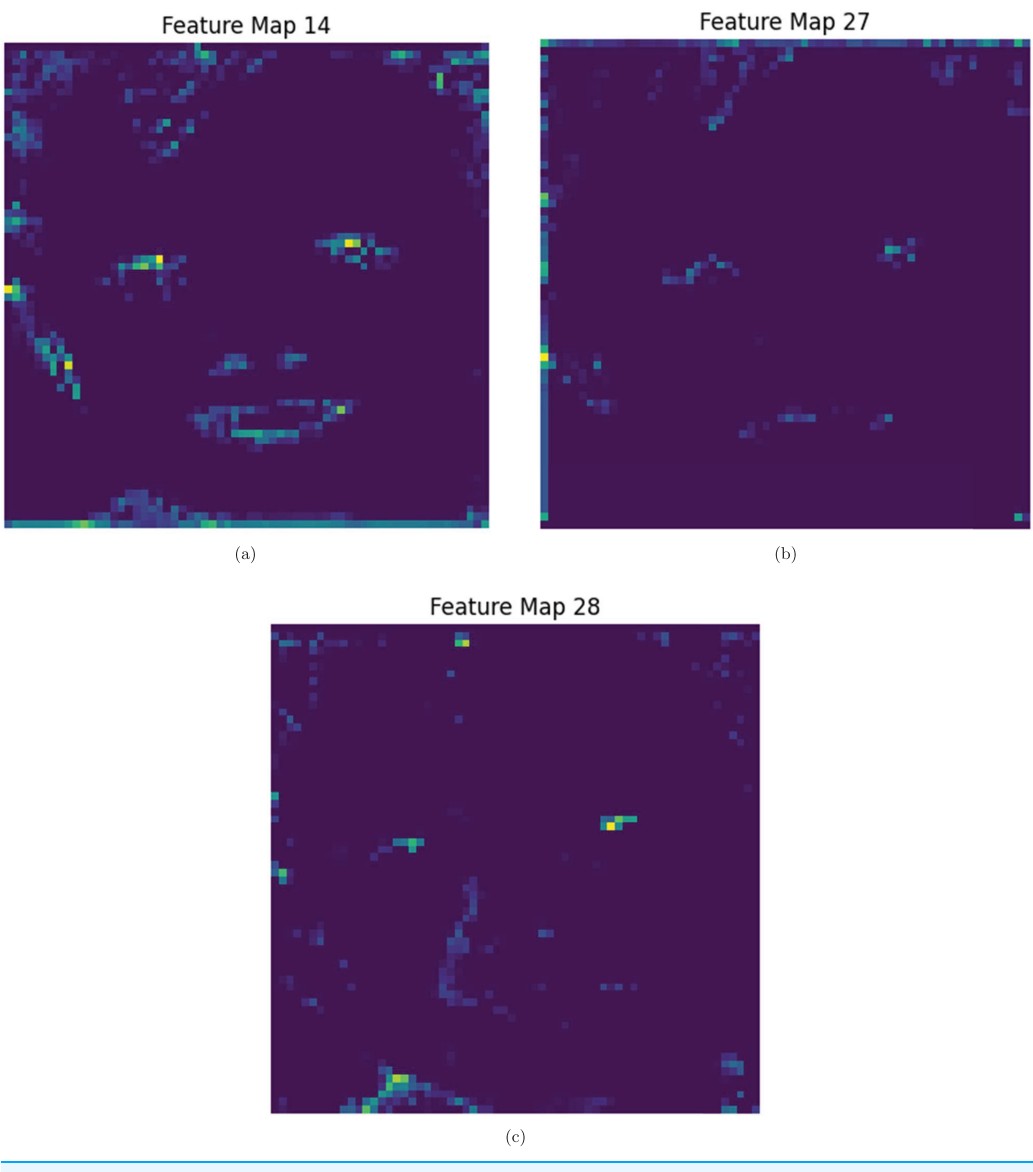

**Figure 12  (A–C) Sample 2: XAI Performance evaluation for Map 14, 27 and 28.**

the diagonal dashed line represents the true positive rate (sensitivity) *vs*. the false positive rate (1-specificity), which serves as a baseline for a random classifier. The results indicate that the multiscale SE-MSF-CNN model outperforms the SE-CNN model, with their ROC curves approaching the top-left corner, signifying higher sensitivity and fewer false positives.

The performance of the optimized models, as illustrated in Figs. 9 and 10, suggests that the WSO-SE-MSF-CNN stands out as the most effective model, demonstrating high accuracy and precision.

From Fig. 9, the WSO-optimized models take longer due to the optimization overhead. However, their performance improvements make the additional training time worthwhile. The SE-MSF-CNN models consistently take longer to train due to the added complexity of

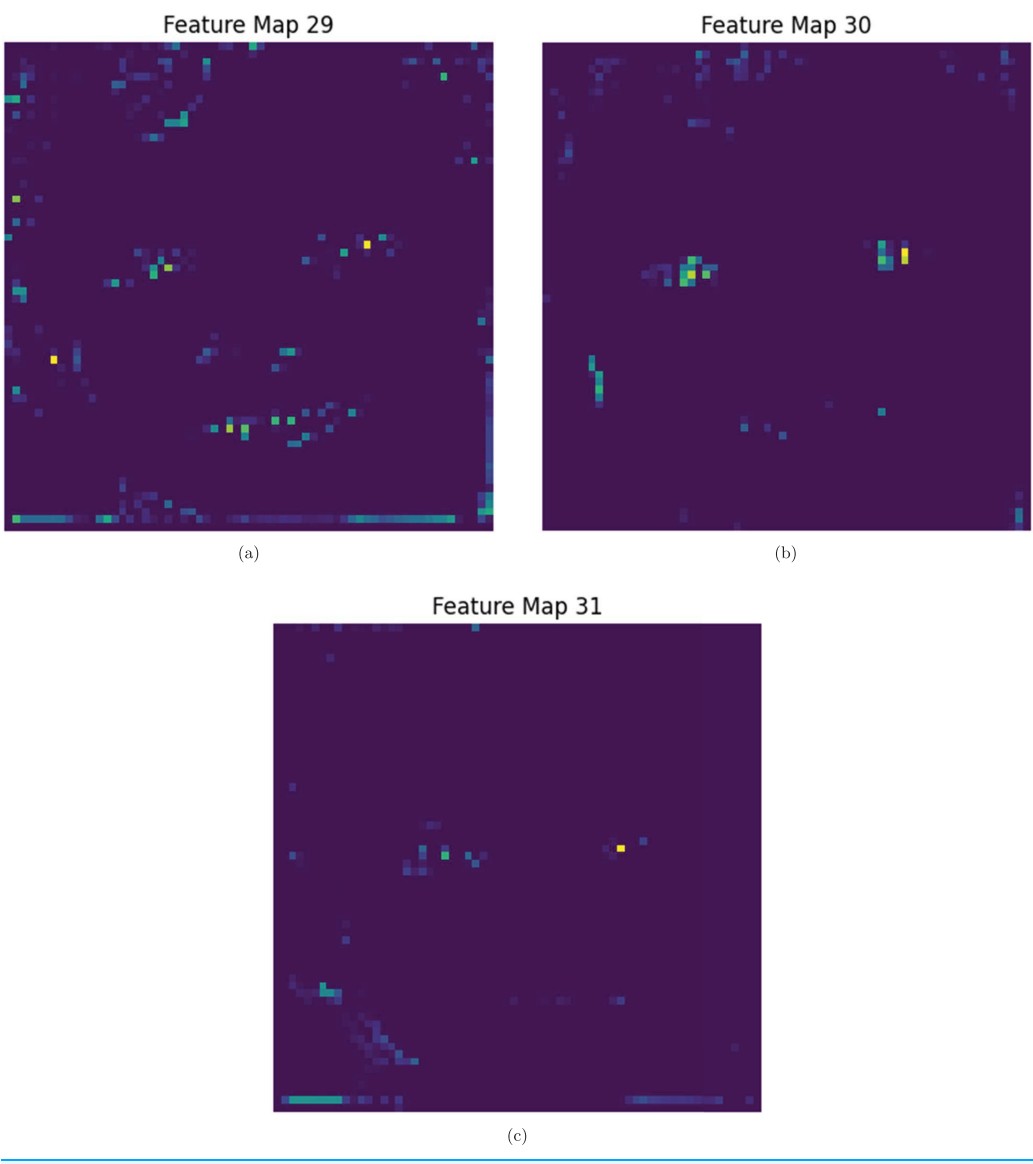

**Figure 13 (A–C) Sample 3: XAI Performance evaluation for Map 29, 30 and 31.**

multiscale feature fusion, but WSO optimization significantly boosts their performance. The improvements from optimization suggest that hyperparameter tuning can significantly enhance model performance, even at the cost of increased training time in some cases. The multi-scale attention-based CNN uncovers prospective patterns associated with autism. By leveraging the attention layers' ability to selectively weigh feature importance, the network can focus on critical areas of facial data (shown in Figs. 11–13), thereby facilitating accurate detection and classification. The deployed architecture also includes dense layers that integrate the extracted features, allowing for a seamless combination of multi-scale attention processes and enhancing training efficiency. The multi-scale attention mechanisms are central to the model's effectiveness, allowing the

**Table 13 Summary of similar state-of-the-art works for autism spectrum disorders detection methods.**

| Ref | ASD classification method | Dataset | Optimization | Result [%] |
|---|---|---|---|---|
| *Cao et al. (2023)* | Vision transformer | | MSE | Acc: 94.5; AUC: 97.9 |
| *Rabbi et al. (2022)* | VGG-19, Inception-V3, DenseNet-201 | | N/A | Acc: 85.0; AUC: 92.3 |
| | | | | Acc: 78.0; AUC: 85.9 |
| | | | | Acc: 83.0; AUC: 91.0 |
| *Alkahtani, Aldhyani & Alzahrani (2023)* | MobileNet and VGG-16 | | Cross entropy | Acc: 92.0; Sen: 92.0; F1: 92.0 |
| | | | | Acc: 82.1; Sen: 82.0; F1: 82.0 |
| *Alam et al. (2022)* | Xception, ResNet-50 | | N/A | Acc: 95.0; AUC: 98.0; Pre: 95.0 |
| | | | | Acc: 94.0; AUC: 96.0; Pre: 94.0 |
| *Mujeeb Rahman & Subashini (2022)* | Xception, EfficientNetB1 | Facial AFID Dataset | CE | Acc: 90.0; Sen: 88.4; Spe: 91.6; AUC: 96.6 |
| | | | | Acc: 89.6; Sen: 86.0; Spe: 94.0; AUC: 95.0 |
| *Akter et al. (2021)* | MobileNet, DenseNet-121 | | N/A | Acc: 92.1 |
| | | | | Acc: 83.6; Sen: 83.6; Spe: 83.6 |
| Proposed model | CNN-based SE | | N/A | Acc: 93.21; Sen: 100; Spe: 88.05 |
| Proposed model | CNN-based-MSF | | N/A | Acc: 94.28; Sen: 96.42; Spe: 92.46 |
| Proposed model | Optimized CNN-based-MSF | | WSO | Acc: 93.21; Sen: 99.29; Spe: 88.54 |
| Proposed model | Optimized CNN-based-MSF | | WSO | Acc: 95.36; Sen: 98.57; Spe: 92.62 |
| *Jeba et al. (2022)* | ResNet-34 | | N/A | Acc: 87.0; Sen: 95; Spe: 82; F1-score: 88.00 |
| *Lakkapragada et al. (2022)* | MobileNetV2-LSTM | | N/A | Acc: 85.0; Sen: 80.4; F1: 84.0 |
| *Ali et al. (2022)* | A multi-modal fusion framework with 3D CNN | Multi-modal SSDB Dataset | N/A | Acc: 75.6; F1: 90.5 |
| Proposed model | CNN-based SE | | N/A | Acc: 89.00; Sen: 86.00; Spe: 91.50 |
| Proposed model | CNN-based-MSF | | N/A | Acc: 85.00; Sen: 82.00; Spe: 87.23 |
| Proposed model | Optimized CNN-based-MSF | | WSO | Acc: 91.00; Sen: 92.00; Spe: 90.19 |
| Proposed model | Optimized CNN-based-MSF | | WSO | Acc: 87.00; Sen: 88.00; Spe: 86.27 |

**Note:**
N/A, Not Available or Applicable; MSE, Mean Square Error; WSO, White Shark Optimization; Acc, Accuracy; Sen, Sensitivity; Spe, Specificity; AUC, Area Under the Curve; F1, F1-score.

network to focus on relevant information at various levels. These methods enhance the network's capacity to capture essential strategies for the early detection and classification of ASD.

# CONCLUSION

One of the most noteworthy aspects of the study is its integration of attention mechanisms within the convolutional neural network design and automated hyperparameter optimization. These mechanisms enable the model to dynamically assess the importance of specific facial regions or features during diagnostic evaluations. The SE-CNN can accurately pinpoint the areas or characteristics crucial for diagnosing ADS in facial images. Clinicians must recognize these facial traits or regions, as they indicate the condition and can help inform their decision-making. Additionally, multi-fusion-based methods

improve diagnostic accuracy and enhance transparency in decision-making. This transparency enables patients and doctors better to understand the rationale behind the model's recommendations, increasing their confidence in its suggestions. A distinguishing feature of WSO is its ability to balance the model's focus on local and global features, ensuring comprehensive contextual awareness while effectively capturing essential fine-grained details. This balance between local and international attention significantly enhances the model's performance, particularly in tasks such as medical image classification, where both the overarching context and specific local features are crucial. Multiscale attention mechanisms have proven essential for capturing complex features in Facial images at both small and large scales, providing a more nuanced understanding of ADS progression. We demonstrate the effectiveness of specific combinations by comparing attention modules and hyperparameter configurations. Among these, the MSF-SE-CNN module combined with the WSO algorithm was particularly effective in detecting ASD with high accuracy. A key limitation of the deployed model is that its deep architecture and multiscale attention layers result in potentially higher computational requirements for deployment, thereby significantly increasing the overall computational costs. This is further compounded by the inclusion of bio-inspired optimization strategies, such as WSO, which also increase the computational burden. While these optimization techniques enhance model performance, they also introduce added complexity and resource-consumption iterative processes, which can compromise scalability and efficiency, particularly in real-time applications. As such, to effectively implement this technology in clinical settings, it is essential to carefully evaluate the trade-off between enhanced performance and the increased resource consumption required for deployment. The SE-MSF-CNN model was also integrated with Grad-CAM to generate heatmaps highlighting facial features relevant to autism classification, particularly regions such as the mouth and eyes. These visualizations align with previous findings, indicating that eye gaze patterns and facial expressions can serve as diagnostic indicators for autism (*Shephard et al., 2020*; *Dawson, Webb & McPartland, 2005*). However, clinical users or experts did not validate our heatmaps, limiting our ability to assess their potential clinical utility. To address this limitation in future work, we recommend collaborating with clinical professionals, such as psychiatrists or autism specialists, to evaluate whether the regions emphasized by Grad-CAM provide meaningful diagnostic information.

## ACKNOWLEDGEMENTS

Generative AI (ChatGPT) has been used to fix English.

### Funding

This work was funded by the Deanship of Scientific Research, Vice Presidency for Graduate Studies and Scientific Research, King Faisal University, Saudi Arabia (Grant No. KFU250488). The funders had no role in study design, data collection and analysis, decision to publish, or preparation of the manuscript.

## Grant Disclosures

The following grant information was disclosed by the authors:
Deanship of Scientific Research, Graduate Studies and Scientific Research, King Faisal University, Saudi Arabia: KFU250488.

## Competing Interests

The authors declare that they have no competing interests.

## Author Contributions

- Walaa N. Ismail conceived and designed the experiments, performed the experiments, analyzed the data, performed the computation work, prepared figures and/or tables, authored or reviewed drafts of the article, and approved the final draft.
- Mona A. S. Ali conceived and designed the experiments, analyzed the data, authored or reviewed drafts of the article, and approved the final draft.

## Data Availability

Code is available at GitHub and Zenodo:

https://github.com/walaanagyis/Autism-Detection/tree/main

walaanagyis. (2025). walaanagyis/Autism-Detection: Autism Spectrum Disorder Detection (V1.0.0). Zenodo. https://doi.org/10.5281/zenodo.16410536

## Supplemental Information

Supplemental information for this article can be found online at http://dx.doi.org/10.7717/peerj-cs.3134#supplemental-information.

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
