# Peer review of "Multiscale attention-based network to enhance detection and classification of autism spectrum disorders using convolutional neural network"

_PeerJ Computer Science, doi:10.7717/peerj-cs.3134_

## Round 0.1 · original submission · Major Revisions

The article has some merit, but the reviewers raised a number of problems that need to be addressed. We kindly ask the authors to prepare an enhanced version of the article addressing all these issues.

**Language Note:** The review process has identified that the English language must be improved. PeerJ can provide language editing services - please contact us at [email protected] for pricing (be sure to provide your manuscript number and title). Alternatively, you should make your own arrangements to improve the language quality and provide details in your response letter. – PeerJ Staff

Reviewer 1 ·

Basic reporting

Introduction and Motivation:
-The introduction is comprehensive but contains redundancy [repeats points like facial features in ASD multiple times] (streamline by removing repetitive sentences and directly stating the motivation once).
Language and Style:
-The manuscript contains typographical and grammatical errors [e.g., "His complexity" instead of "This complexity", "Radio-facial" instead of "Cranio-facial"] (perform a thorough professional proofreading and English editing pass).
Formal Presentation of Results:
-Definitions of technical components like WSO optimization and fitness function are missing or insufficient [missing detailed math formulas for fitness or loss function] (add formal definitions and equations where needed for clarity and rigor).
Proofs and Algorithmic Details:
-The WSO algorithm is described narratively without clear pseudocode [steps described in text but no pseudocode or structured listing] (include a pseudocode block or structured flowchart summarizing WSO steps).
Structure and PeerJ Standards Compliance:
-While the article is detailed, figures (e.g., Figures 13–15) are referenced without full explanation [figures inserted but not fully discussed when first introduced] (explicitly describe every figure and its relevance in the text where it first appears).

Experimental design

The methodology is described with sufficient detail to replicate the study [code submission is confirmed; datasets (Piosenka and YTUIA) are cited and described; hyperparameter settings, WSO steps, and model structures (SE-MSF-CNN) are explained thoroughly].

Data preprocessing steps—such as face cropping, scaling, alignment using MTCNN, and normalization—are included and are appropriate for the problem context [sufficient discussion is provided on cleaning and preparing data from online and video sources].

Evaluation metrics (accuracy, precision, recall, F1-score, specificity, and Cohen’s Kappa) are adequately described and used correctly, but statistical significance of the results is not discussed [suggestion: include standard deviations or statistical testing across runs to validate metric stability].

Model selection methods are indirectly described through ablation studies and performance comparisons across configurations (e.g., with/without WSO or SE blocks), but could be strengthened [suggestion: clarify how final model versions were selected, e.g., based on validation loss, AUC, or early stopping criteria].

Validity of the findings

Impact and Novelty:
The article proposes a novel integration of Squeeze-and-Excitation Networks (SE-CNN), Multiscale Feature Attention (SE-MSF-CNN), and White Shark Optimization (WSO) for ASD detection.
Conclusions:
The conclusions summarize results based on SE-MSF-CNN and WSO-enhanced performance but overstate clinical impact. Suggestion: restrict conclusions strictly to model performance on datasets and recommend separate clinical validation for patient-level outcomes.

Experimental Quality:
The experiments utilize datasets (Piosenka, YTUIA), SE-CNN, SE-MSF-CNN, and optimization through WSO with evaluation metrics like accuracy, precision, recall, F1-score, Cohen's Kappa, but lack statistical significance testing [no p-values, confidence intervals, or standard deviations are reported for comparisons across models (e.g., SE-CNN vs SE-MSF-CNN with/without WSO); suggestion: perform statistical significance tests like paired t-tests or McNemar’s test between baseline and optimized models to validate the results].

Conclusion Identification:
The conclusion only discusses performance improvements but does not recognize model limitations such as computational overhead from multiscale attention or WSO [no mention of limitations like increased training time (noted in computational analysis Figure 11) or challenges in real-time clinical use; suggestion: add a paragraph on limitations like high computational cost and propose lightweight versions or model compression for future work].

Additional comments

The manuscript presents a technically sound and timely contribution to ASD detection using deep learning; however, the manuscript would benefit from professional copyediting to improve grammar, eliminate typographical errors (e.g., “His complexity” instead of “This complexity”), and ensure consistent formatting across sections, figures, and citations. Additionally, figure captions (especially for Figures 13–15) could be made more informative, and visualizations like Grad-CAM heatmaps would be more useful if paired with clinical interpretation or expert annotation to reinforce their diagnostic value.

Reviewer 2 ·

Basic reporting

(1)
The manuscript is generally well-structured and written in clear English, though some minor grammatical issues suggest that further language polishing is needed.

(2)
Additionally, the paper does not provide access to source code or pretrained models, limiting reproducibility. Authors could deposit their code in a public repository and provide links in the manuscript.

(3)
The literature review demonstrates a commendable effort. However, it would be valuable to expand the discussion on non-invasive methods coupled with Machine Learning for autism diagnosis, such as eye-tracking. Especially, studies that applied that applied Transfer Learning or Attention in that context.

Experimental design

While the proposed approach is innovative, there are several design issues that need addressing:
(1)
The paper uses facial image datasets sourced from online platforms, which likely contain multiple images of the same individuals. It is unclear whether the data were split by subject or by image. If not done at the subject level, this could lead to information leakage and inflate performance metrics.

(2)
The model’s evaluation is limited to two internally partitioned datasets. Without testing on an external dataset or through cross-dataset validation, the generalizability of the approach is unproven.

(3)
The White Shark Optimization algorithm is introduced without comparison to more commonly used hyperparameter optimization methods (e.g., grid search, Bayesian optimization). Its advantage remains speculative without such baseline comparisons.

Validity of the findings

(1)
The reported performance metrics are (unusually) high for a task involving real-world facial image data, raising concerns about overfitting or potential information leakage. The lack of subject-level data splitting, external validation, or data de-duplication checks undermines the reliability of these findings.

(2)
While the Grad-CAM visualizations offer interpretability, no user or expert validation is performed to assess their clinical utility.

(3)
Ethical concerns, including dataset bias, privacy, and real-world deployment constraints, are not sufficiently addressed.

Additional comments

Overall, I appreciate the authors' efforts and look forward to seeing an improved version of this study.

---

## Round 0.2 · Major Revisions

The reviewer again found some problems and drawbacks regarding the study. I invite again the reviewers to consider them and to prepare an enhanced version of the article.

**PeerJ Staff Note**: Please ensure that all review, editorial, and staff comments are addressed in a response letter and that any edits or clarifications mentioned in the letter are also inserted into the revised manuscript where appropriate.

Reviewer 2 ·

Basic reporting

I appreciate the authors’ responses, and I would like to thank them again for their efforts. However, I am afraid that there are methodological concerns that would require further attention.

(1)
Regarding information leakage, the authors’ use of a hash check partially addresses duplicate image leakage but does not fully eliminate the risk of subject-level leakage, in my view. Perceptual hashing detects ‘near-identical’ pixel-level copies but does not account for different images of the same subject, which can be highly correlated. For video-derived datasets, splitting should be performed at the subject or video level, not at the frame level, to prevent temporal leakage.

(2)
The study still doesn’t provide a convincing rationale or evidence behind the choice of WSO. The accuracy difference between WSO and Bayesian is only ~1% — within the standard deviation (±0.02). Their confidence intervals overlap! So statistically, it’s unclear if there’s a true performance difference.

(3)
The time comparisons reported in Table 8 show that WSO required ~1 hour, Bayesian optimisation ~1.5 hours, and grid search ~2 hours — but the number of hyperparameter evaluations, total search space, and stopping criteria for each method are not fully detailed. For a fair comparison of computational efficiency, all methods should be run under equivalent computational conditions.

Furthermore, grid search is naturally exhaustive by design, so it is expected to be slower but guarantees coverage of the search space. Bayesian is known to be more efficient than grid search for small search spaces, so the fact that WSO’s time difference is only ~30 minutes faster than Bayesian optimisation for a (small) 2-parameter problem is not sufficient to claim clear computational advantage, in my view.

(4)
I encourage the authors to deposit the code and models in a public repository with proper documentation. Or to justify clearly why this is not possible.

Experimental design

N/A

Validity of the findings

N/A

Additional comments

N/A

---

## Round 0.3 · accepted · Accept

The reviewer is satisfied with the recent changes to the article and therefore I can recommend this article for acceptance.

Reviewer 1 ·

Basic reporting

The author has modified according to the given comments.

Experimental design

The author has modified according to the given comments.

Validity of the findings

The author has modified according to the given comments.

Additional comments

The author has modified according to the given comments.